# Thermochemiluminescent peroxide crystals

Stefan Schramm [1], Durga Prasad Karothu [1], Nathan M. Lui [1], Patrick Commins [1], Ejaz Ahmed [1], Luca Catalano [1], Liang Li [1], James Weston[1], Taro Moriwaki[2], Kyril M. Solntsev [1,3] & Panče Naumov [1,4]

Chemiluminescence, a process of transduction of energy stored within chemical bonds of ground-state reactants into light via high-energy excited intermediates, is known in solution, but has remained undetected in macroscopic crystalline solids. By detecting thermally induced chemiluminescence from centimeter-size crystals of an organic peroxide here we demonstrate direct transduction of heat into light by thermochemiluminescence of bulk crystals. Heating of crystals of lophine hydroperoxide to ~115 °C results in detectable emission of blue-green light with maximum at 530 nm with low chemiluminescent quantum yield [$(2.1 \pm 0.1) \times 10^{-7}$ E mol$^{-1}$]. Spectral comparison of the thermochemiluminescence in the solid state and in solution revealed that the solid-state thermochemiluminescence of lophine peroxide is due to emission from deprotonated lophine. With selected 1,2-dioxetane, endo-peroxide and aroyl peroxide we also establish that the thermochemiluminescence is common for crystalline peroxides, with the color of the emitted light varying from blue to green to red.

[1] New York University Abu Dhabi, P.O. Box 129188, Abu Dhabi, UAE. [2] Japan Synchrotron Radiation Research Institute, 1-1-1 Kouto, Sayo, Hyogo 679-5198, Japan. [3] School of Chemistry and Biochemistry, Georgia Institute of Technology, Atlanta, GA 30332-0400, USA. [4] Radcliffe Institute for Advanced Study, Harvard University, 10 Garden St, Cambridge, MA 02138, USA. Correspondence and requests for materials should be addressed to P.N. (email: pance.naumov@nyu.edu)

Emission of light by chemiluminescence is due to a chemical reaction between reactants in the ground state that results in an unstable high-energy intermediate. Strongly exothermic decomposition of the intermediate subsequently affords products in their electronically excited state, followed by emission of visible light. When this light is generated by a living organism, in many cases by catalysis in an enzymatic scaffold, the process is known as bioluminescence, and is used by lower biological organisms to communicate, attract prey, or mate[1]. The chemiluminescence is also at the core of the generation of 'cold light' used in applications that range from glow sticks to single molecule detection and ultra-sensitive tumor bioimaging. The high-energy reaction intermediates are usually thermally labile hydroperoxides, 1,2-dioxetanes, 1,2-dioxetaneones (1,2-dioxetanones), or 1,2-dioxetanedione[2–4]. If these chemiluminophores are chemically modified to be stable at room temperature, their decomposition can be induced by heating or by application of mechanical force, and the respective phenomena are referred to as thermo- and mechanochemiluminescence. It was demonstrated recently that mechanochemiluminescent polymers can be prepared by incorporating mechanoluminophores in their polymer backbone[5,6], and thermochemiluminescence was also accomplished by heating of deposited nanoparticles[7–9] and microcrystalline powder[10]. Chemiluminescence in the pure macroscopic solid state, however, was deemed unfeasible due to the very limited ability of molecules for diffusion required for their reaction and the anticipated low quantum yields.

In search for a material that would display thermochemiluminescence in crystalline state, we turned our attention to lophine (2,4,5-triphenyl-1*H*-imidazole), the first reported organic chemiluminescent molecule[11], which from the extensive studies by Kumura, White, and others is known to emit light upon dissolution in basic alcoholic solvents[12–29]. Its hydroperoxide (LHP; Fig. 1a) is stable at room temperature in the solid state, but it decomposes to lophine and dibenzoylamidine at high temperatures[16]. As the light emission by thermochemiluminescence is triggered only by heat and does not require chemical reagents, it is foreseen as a convenient analytical tool that combines high signal-to-noise ratio, ultrasensitivity required for detection in small sample volumes, and technically simple instrumentation.

Here we report evidence of emission of light from macroscopic crystals of organic peroxide, and we provide the spectral profile and mechanism of its thermochemiluminescence. We also demonstrate that the solid-state thermochemiluminescence is not limited to LHP but is common for cyclic, endo, and aroyl peroxides, and the color and spectrum of the emitted light are specific to the respective decomposition products. These results open prospects for application of organic peroxides as multicolor thermally sensitive solid light-emitting materials.

## Results

**Synthesis and thermochemiluminescence**. LHP was synthesized by Schenck-ene photooxygenation of lophine at −10 °C in presence of methylene blue adsorbed onto silica as a sensitizer in a custom-built LED photoreactor (for details on the synthesis, characterization and experimental setup, see Supplementary Methods). The product was crystallized from several organic solvents. Although thermogravimetric analysis indicated that there is no significant decomposition at room temperature, for safety reasons the crystals were stored below −20 °C. As shown in Fig. 1b, c, when crystals of LHP are heated to their decomposition temperature (116.5 °C, Fig. 1d) they decompose and emit visible blue-green light with emission maximum at 530 nm (CIE (International Commission on Illumination) 1931 color space coordinates: $x = 0.2854$, $y = 0.4009$). The light emission

decays over a period of several minutes. LHP can be obtained in centimeter-size crystalline aggregates, and the spatial progression of the reaction can be observed from the shift in the light emission through large crystals. Crystals of LHP were heated to 120 °C on a heating stage and their light emission was recorded by using a thermoelectric cooled CMOS camera attached to a low-light microscopic setup[30,31] (details are provided as Supplementary Methods). Supplementary Movies 1–3 show the light emission as recorded from the top of the crystals as they were heated from the bottom. The light emission starts beneath the crystals at their contact point with the heated stage and progresses throughout their bulk, and can be seen as a reaction front moving across the temperature gradient. The crystals gradually light up, and after the reaction concludes, the light emission ceases.

To visualize the spatial progress of the reaction, a 15 mm-crystal agglomerate weighing 690 mg was heated while observing it from the side (Fig. 1d, Supplementary Movie 4). The front of the emitted light progressed through the crystal and the emission ceased at the region where the thermal convection was insufficient to reach the decomposition temperature, leaving both reacted and non-reacted domains in the partially reacted crystal. The light emitted from the interior of the crystal is partially filtered by its exterior, resulting in gradual change of the emitted color from cyan to dark yellow to orange as the reaction progresses throughout the crystal (Fig. 1d, Supplementary Movie 4). The reacted domain was visually observable as discoloration on the surface of the partially decomposed crystal that was in direct contact with the heater (Figure 1f). 3D visualization of the reacted crystal by computed tomography (CT) provided the exact morphology of the reacted and unreacted parts as domains of different densities in the crystal interior (Fig. 1e, g–k). The shape of the interface between the two domains indicates that the reaction has advanced radially from the contact surface and throughout the bulk of the crystal, and ceased 4.4 mm from the heated base due to drop of the temperature below the decomposition point across the temperature gradient.

When the heating is performed on crystals submerged in a drop of oil, evolution of oxygen is observed around 110 °C (Fig. 1l–o). The decomposition is accompanied by violent disintegration of the crystals (Supplementary Movie 6). Variable-temperature powder X-ray diffraction was used to study the thermal decomposition of LHP (Fig. 2a). The starting material retains its structure up to about 100 °C. The reaction results in changes in the powder X-ray diffraction pattern between 100 and 110 °C. The product of the reaction was further heated until 150 °C and did not show additional changes. This result is in line with the results from the thermal analysis (Fig. 1c) and confirms that the major structural changes in the crystal happen in the temperature range 100–110 °C. In order to study the isothermal kinetics of the associated chemical reaction, μIR spectroscopy using synchrotron IR radiation focused on a single crystal of LHP maintained at 115 °C was applied (Fig. 2c). As inferred by comparison of the spectra before and after heating for 1120 s, the reaction resulted in significant changes in the IR spectrum (Fig. 2b). Evolution of at least seven characteristic bands with average kinetic constant of $(3.0 \pm 0.8) \times 10^{-3}$ s$^{-1}$ was observed in the region 1706–915 cm$^{-1}$ (Fig. 2d, e). DFT calculations (for details, see the Supplementary Methods) identified lophine as the main reaction product, and this result is in accord with the release of oxygen during decomposition. Lophine was also isolated from the reacted crystals and identified as the main product by using NMR spectroscopy, mass spectrometry and structure determination by single crystal X-ray diffraction. Analytical UHPLC-MS indicated formation of two side products in <5% yield, one of

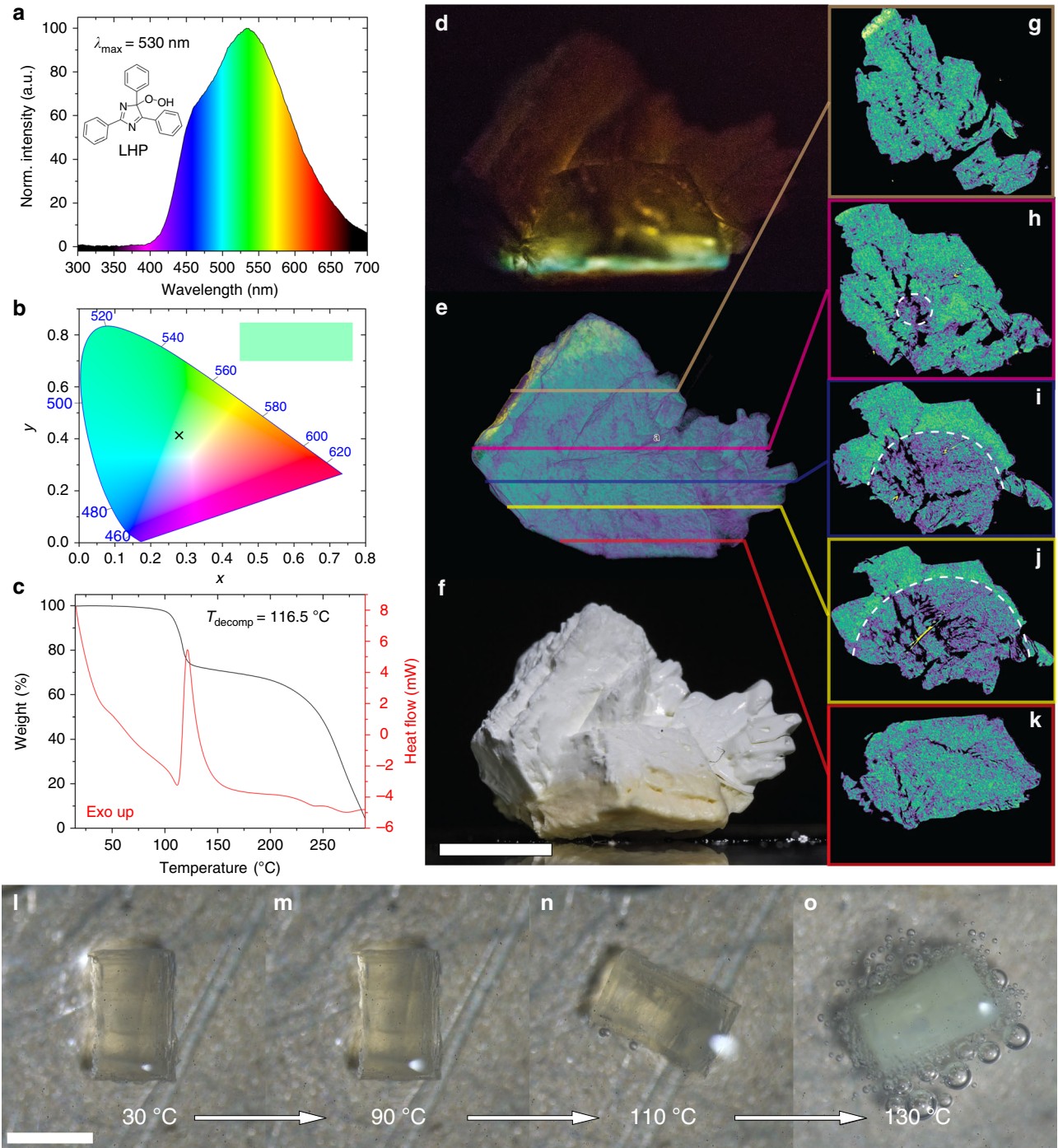

**Fig. 1** Thermochemiluminescence and decomposition of LHP. **a** Molecular formula and solid-state thermochemiluminescence spectrum of LHP ($\lambda_{max}$ is the wavelength of maximum emission). **b** Color of the light emission plotted on a CIE (International Commission on Illumination) 1931 color space diagram showing the position of the thermochemiluminescence of LHP with coordinates $x = 0.2854$ and $y = 0.4009$. **c** Thermogravimetry of solid LHP showing its decomposition at 116.5 °C ($T_{decomp}$ stands for decomposition temperature). **d** Thermochemiluminescence of a crystal agglomerate of LHP heated from the bottom (recording of the light emission is available as Supplementary Movie 4). The difference in color of the emitted light at different locations is caused by direct emission and reabsorption/filtering of the emitted light through the interior of the crystal. **e** Computed tomography (CT) scan of the partially reacted crystal. **f** Bright-field image of the same crystal after the reaction. The reacted part of the crystal is seen as discolored section. **g–k** 2D slices taken at varying depth through the 3D CT scan showing different densities in the reacted and non-reacted domains of the crystal (the 3D scan is available as Supplementary Movie 5). **l–o** A crystal of LHP heated in a drop of oil to 110 °C starts to decompose by release of oxygen, as seen by the evolution of bubbles. The length of the scale bar in **f** (also applies to **d** and **e**) is 1 cm. The length of the scale bar in **l** (also applies to **m–o**) is ~1 mm

which was isolated and identified with X-ray diffraction and other methods as dibenzoylamidine, a known hydrolysis product of lophine[12,26]. The formation of trace of an imidazolone as side-product has been reported for the stereoselective thermal

rearrangement of chiral lophine peroxides, in particular silyl derivatives, in solution[24], however, we could not observe this product by thermally induced chemiluminescence in any of our experiments.

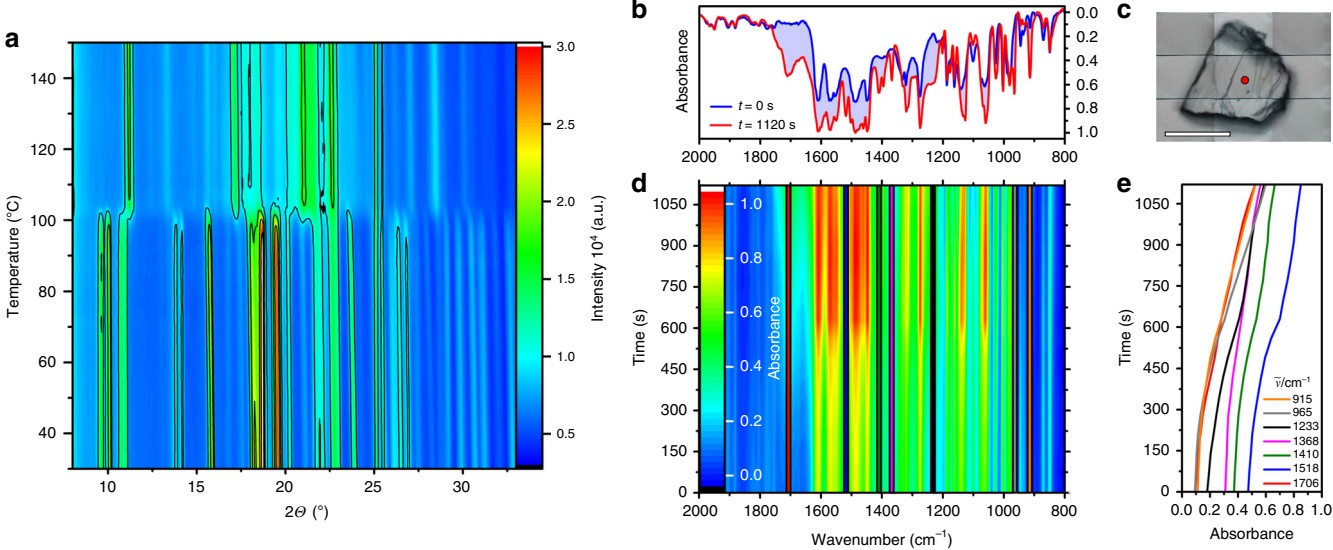

**Fig. 2** μIR spectra and powder XRD analysis of decomposition of LHP. **a** Variable-temperature powder X-ray diffraction pattern of LHP heated from room temperature to 150 °C. **b** The 2000–800 cm⁻¹ region in the IR spectrum before (blue line) and after heating to 115 °C for 1120 s (red line). **c** Optical image of the LHP crystal used to record the μIR spectrum (the red circle indicates the position of the incident IR beam). The scale bar corresponds to 50 μm. **d** Time-profile of the IR spectrum of LHP at 115 °C. **e** Isothermal kinetic traces at selected wavenumbers (highlighted in **d**) used in the kinetic analysis

**Spectroscopic characterization**. To gain insight into the origin of the emitted light, the thermochemiluminescence of LHP (maximum around 530 nm) was analyzed by a combination of spectroscopic methods. The time-dependent thermochemiluminescence two-dimensional excitation-emission spectrum of LHP in the solid state (Fig. 3a) does not display changes in the emitted color and points out to a single emitting species or multiple species at fast equilibrium. The chemiluminescence quantum yield, determined by using absolute calibration methods, was $(2.1 \pm 0.1) \times 10^{-7}$ E mol⁻¹ ($n = 10$), and thus it is close to the yield reported for the reaction in solution[13]. The very low quantum yield is inherent to the multiple degrees of rotational freedom of the emitter that open channels for radiationless decay. The activation parameters extracted from solid-state kinetics measurements (Fig. 3b) are higher than the values for the solution reaction in non-basic solvent (chlorobenzene (CB)) and polar basic solvent (N-methyl-2-pyrrolidone (NMP)), indicating that the emission involves two processes that counteract each other. The higher activation parameters in the solid state indicate that the lattice of the crystal increases the energy required for decomposition, while the stabilization by polar solvents points out to a highly polar transition state. Moreover, the activation entropy in the solid is positive, while it is negative in both solvents. Therefore, the transition state in the solid is less ordered in the crystal, while a certain reorganization occurs in solvents, particularly in NMP. This observation is consistent with the polar nature of the proposed transition state (Fig. 4a).

Contrary to the solid-state, the thermochemiluminescence spectra of LHP recorded in solvents with polarity ranging from non-polar and non-basic (dodecane, $\varepsilon = 2$) to very polar and basic medium (dimethylformamide, $\varepsilon = 36.7$; Fig. 3c) reveal two emitting centers, with emissions around 430 and 530 nm that are very weakly affected by the solvent. In most cases, the two close-lying emissions merge into a broad band with a shoulder that resembles the solid-state thermochemiluminescence spectrum. In a Kamlet-Taft analysis[32,33] of the solvent effects, the 430 nm band did not display a reliable statistical significance, while the 530 nm band underwent a small bathochromic shift with decreasing solvent polarity and increasing solvent basicity. Surprisingly, the $I_{430}/I_{530}$ intensity ratio increased with increasing solvent polarity

and decreasing solvent basicity, revealing that the species at 530 nm is the main emission in basic solvents, while the species at 430 nm becomes dominant in non-basic solvents. This result points out to simultaneous emission from the singlet and triplet excited states of the same species in solution, and the ratio of the two emissions depends upon its microenvironment.

While some peroxides, and particularly dioxetanes, are known to emit from a triplet state upon thermal decomposition[2,34,35], other chemiluminescent molecules such as 2-coumaranones[36–39] emit mainly from their excited singlet state. When the molecule is excited by light, emission from singlet state is equivalent to fluorescence, while emission from a triplet state results in phosphorescence. Many of the organic metal-free room-temperature phosphorescent systems developed recently[40–42] exhibit dual emission (fluorescence and phosphorescence) from the same molecule. To obtain further insight into the origin of the thermally induced emission from solid LHP, we studied the photophysical properties of its main reaction product (>95% yield), lophine. Pure solid lophine shows fluorescence at 383 nm and phosphorescence at 542 nm (Fig. 3d). While the phosphorescence emission of lophine is close to the solid-state thermochemiluminescence of LHP (530 nm), its fluorescence is far from the LHP emission in solution (430 nm). Therefore, the photophysical properties of lophine can not explain the observed solid-state thermochemiluminescence emission spectra, and necessitated deeper insight into the lophine photochemistry. Consequently, photoluminescence spectra in solidified neutral, basic and acidic matrices of CB and NMP were recorded at −196 °C (Fig. 3e, f). The emission spectra recorded in neutral conditions resemble the solid-state spectra. Contrary to this, the spectra recorded in acidic conditions are very different between CB and NMP. While the spectrum in CB shows a complex fluorescence emission with multiple maxima between 450 and 510 nm, the fluorescence maximum in acidic NMP lies very close to the one in neutral medium (ca. 380 nm). The results in basic media are especially striking, and in both solvents lophine shows fluorescence emission around 430 nm and phosphorescence emission around 530 nm. Based on the phosphorescence emission of lophine in basic solid matrices (530 nm) and the

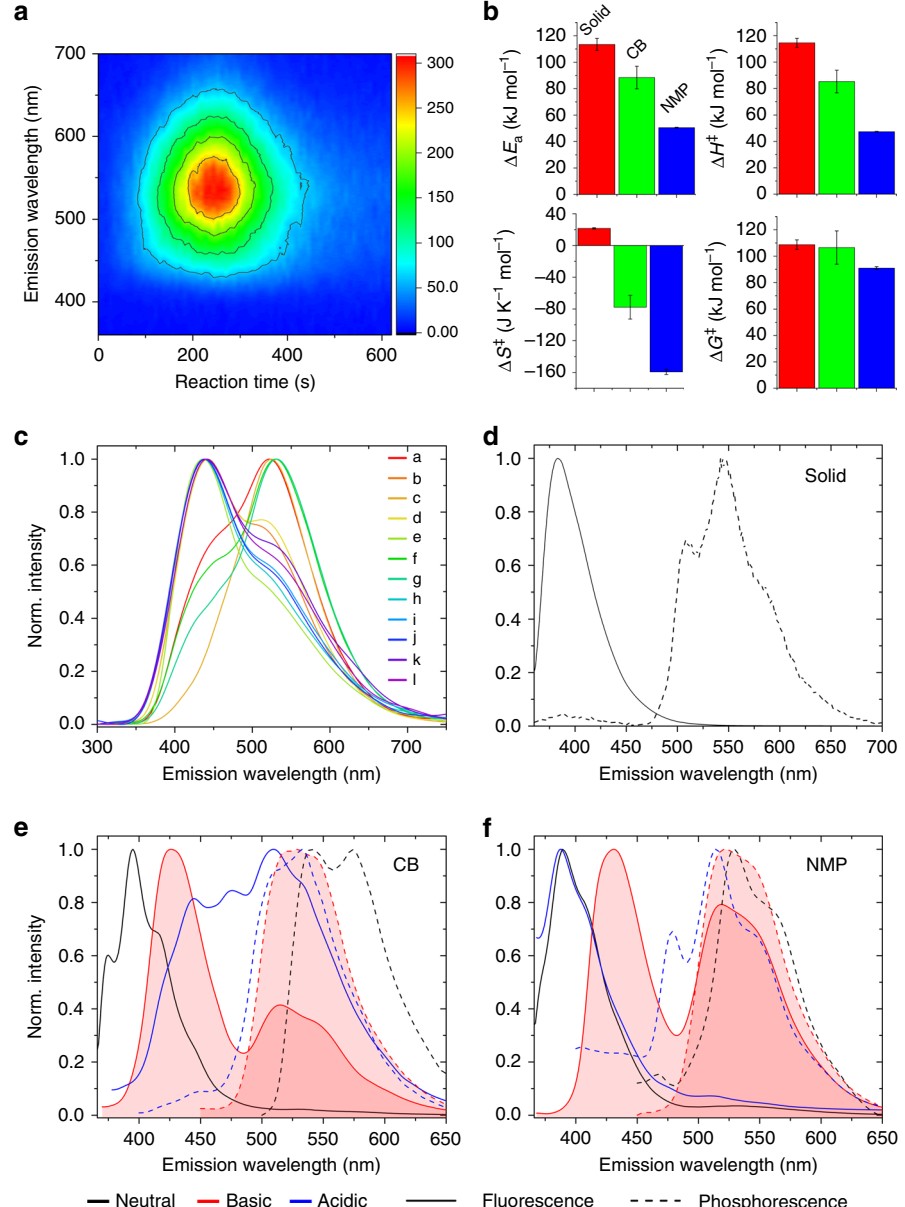

**Fig. 3** Spectroscopic characterization of LHP thermochemiluminescence. **a** Evolution of the thermochemiluminescence spectrum of LHP over time showing absence of change in the emission maximum throughout the reaction. The color code scale given to the right corresponds to emission intensity in arbitrary units. **b** Activation parameters of the thermochemiluminescence of LHP in the solid state and in solutions of chlorobenzene (CB) and N-methyl-2-pyrrolidone (NMP) as typical non-polar and polar solvents. The error bars represent standard deviations. **c** Thermochemiluminescence spectra of LHP in 12 solvents ranging from non-polar to polar: dimethylformamide (a), dimethylacetamide (b), N-methyl-2-pyrrolidone (c), ethylene glycol (d), acetophenone (e), 1-hexanol (f), 1-octanol (g), chlorobenzene (h), 1,2,4-trimethylbenzene (i), toluene (j), hexadecane (k), and dodecane (l). **d** Fluorescence (room temperature) and phosphorescence (–196 °C) spectra of the main reaction product, lophine, recorded as solid. (**e**, **f**) Fluorescence and phosphorescence spectra of lophine recorded in neutral, acidic, and basic matrices at –196 °C in CB (**e**) and in NMP (**f**)

thermochemiluminescence of LHP in basic solvents and in the solid state (530 nm), and taking into account that the maxima of the thermochemiluminescence reaction in the solid state does not change over time (Fig. 3a) and that its kinetics is of first order (Supplementary Figure 15), we hypothesized that the actual emitter in the solid-state thermochemiluminescence of LHP is the excited triplet state of deprotonated lophine. The triplet emission necessitates intersystem crossing. Such an intersystem crossing is forbidden by the quantum mechanical principles. Only a very small amount of molecules will actually cross to their excited triplet state and even less will emit from this state. Hence, the

observed thermochemiluminescent quantum yield is inherently very low. The solvent dependence of this binary system ($S_1$ or $T_1$ emission) opens prospects for its future use as ratiometric dye. Such tuning of the dual emission is relevant to the development of long-lived emissive probes for time-resolved photoluminescence bioimaging and biosensing, and could also be applied to multiplex immunoassays. We note that during the experiments we did not encounter any indication of intense reddish-purple coloration that is common for the lophine radical and would indicate homolytic bond cleavage, neither did we observe any lophine radicals or their dimers in the UHPLC-MS experiments

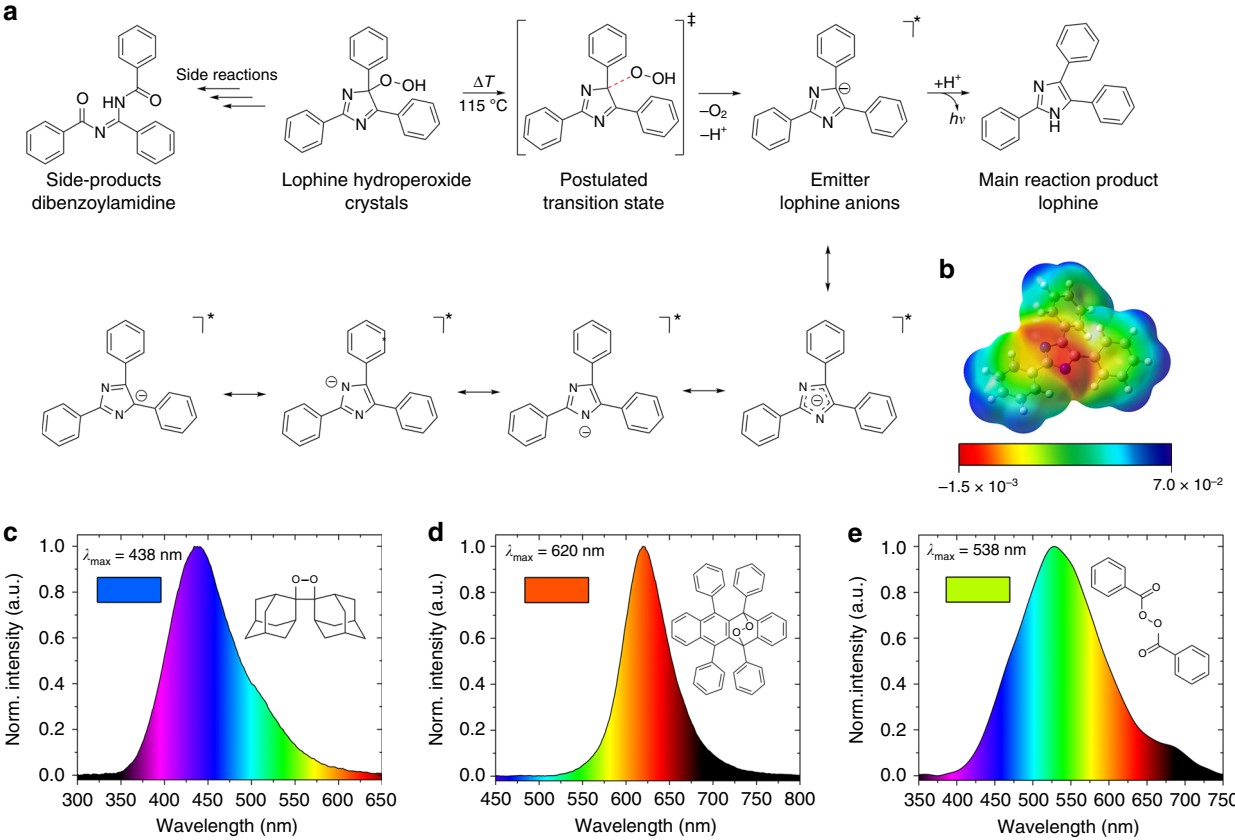

**Fig. 4** Thermochemiluminesence and spectra of LHP and other peroxides. **a** Proposed reaction mechanism for the solid-state thermochemiluminescence of LHP. **b** Electrostatic potential isosurface of the lophine anion mapped onto its total density surface, showing delocalization of the negative charge across the central imidazole core. The color-coded scale is given on the bottom with the potential in relative units, increasing from red (most negative) to blue (most positive). **c–e** Solid-state thermochemiluminescence spectra of other organic peroxides selected as representative members of different peroxide classes: *bis*(adamantly)-1,2-dioxetane (1,2-dioxetane) (**c**), rubrene endoperoxide (endoperoxide) (**d**), and benzoyl peroxide (aroyl peroxide) (**e**)

(Supplementary Figure 3). Therefore, we conclude that radicals are not formed under our experimental conditions and we exclude the possibility for a homolytic bond cleavage.

**Thermochemiluminescence mechanism and emission from other peroxides**. Based on the results, we posit a reaction mechanism for the LHP solid-state thermochemiluminescence as outlined in Fig. 4a. Heating of the crystalline hydroperoxide to its decomposition temperature, around 115 °C, results in exothermic breaking of the C–O bond on the peroxy moiety and affords the excited state of deprotonated lophine. As it was confirmed by quantum chemical calculations, which show that the charge is distributed across the central core (Fig. 4b), as well as with the calculated IBO charges (for details on the computations, see the Supplementary Methods), the lophine anion is strongly stabilized by distribution of the negative charge throughout the imidazole core across its multiple resonance structures. The excited lophine anion subsequently relaxes to the ground state. We hypothesize that the side-product (dibenzoylamidine) might be formed by a parallel reaction. In that scenario, initiated by the basic lophine anion that is formed in the main thermochemiluminescence reaction, LHP is converted to the corresponding 1,2-dioxetane after cyclization, and the product decomposes through O–O and C–C bonds cleavage to generate dibenzoylamidine. This hypothesis is consistent with the base-induced decomposition of LHP reported by White et al.[13,14].

To verify whether the solid-state thermochemiluminescence is limited to LHP or is a more common phenomenon, the emission from other organic peroxide classes, namely 1,2-dioxetanes, endoperoxides and aroyl peroxides in form of macroscopic crystals, was also recorded. Depending on the chemical substituents and molecular weight, peroxides can be highly explosive, and therefore one representative compound was selected as a representative of each compound class that has high reported decomposition temperature and is stable as solid at ambient conditions. The *bis*(adamantly)-1,2-dioxetane was selected as the most stable 1,2-dioxetane. The choice from endoperoxides was set on rubrene endoperoxide, a photooxidation product of rubrene that is commonly used in photovoltaics, and the commercially available polymerization initiator benzoyl peroxide was selected as an example of the aroyl peroxides. When heated to their respective decomposition temperature (between 105 and 160 °C), each of these solid compounds immediately produced chemiluminescence signal that was recorded spectroscopically. As shown in Figure 4c–e, *bis*(adamantly)-1,2-dioxetane emits blue light with emission maximum of 438 nm with rather slow kinetics, benzoyl peroxide emits in the green region with a maximum of 538 nm, and rubrene endoperoxide emits red light with a maximum at 620 nm. This result does not only demonstrate that the solid-state thermochemiluminescence is common and detectable for other peroxides and possibly other high-energy molecules, but it also shows that the thermally induced emission depends on the temperature and by combining different thermochemilumino-phores it can cover the entire visible spectrum, a property that could have implications for future sensing applications based on solid-state thermochemiluminescence.

**Conclusion**. In summary, on the example of lophine hydroperoxide, we have provided evidence of thermochemiluminescence from macroscopic-sized organic crystals. Analysis of the reaction mechanism confirmed the generation of two emitting centers of different spin multiplicity and radiative lifetimes in solution, a property that favors this material as ratiometric luminescence dye. For the peroxide class of molecules, a built-in self-calibration allows for a precise determination of the microenvironment parameters. We have shown that fine emission color tuning can be achieved by changing the solvent, temperature, and time-gating of the experiment. The very low quantum yield of the solid-state thermochemiluminescence is of the same order as that observed in solution and is inherent to the emission from the triplet state and the molecular flexibility of the emitter. This observation of thermochemiluminescence from organic solid-state sets the path to the development of other, more efficient solid chemiluminescent systems, for example, by chemical modification aimed at optimization of the precursor structure to enhance the light output from the reaction product. We have also demonstrated that the light emission induced by thermal decomposition of high-energy crystals is common for other classes of organic peroxides and possibly for other chemiluminophores, and although it might occur through different mechanisms and lead to different products depending on the reactant, they all emit detectable visible light. This phenomenon opens up an unexplored direction in solid-state chemiluminescence research with potentials for possible applications in both materials research and life sciences.

## Methods

**Synthesis and product isolation**. LHP was synthesized by photooxygenation via a Schenk-ene reaction from lophine (2,4,5-triphenyl-1H-imidazole) and singlet oxygen in a custom-built RGB photoreactor using the photosensitizer methylene blue. Further details on the design and technical characteristics are provided as Supplementary Methods.

**Spectroscopy**. Variable-temperature μIR spectroscopic experiments were performed at beamline BL43IR at SPring-8, Japan. A Vertex70 FTIR spectrometer equipped with a Bruker Hyperion 2000 infrared microscope and MCT detector was used. All chemiluminescence, fluorescence and phosphorescence spectra or kinetics were recorded on a Jasco FP8500 spectrofluorimeter equipped with either an ILFC-847S-cooled integration sphere and ESC-842 reference light source for the quantum yield and low-temperature measurements or a ETC-815 temperature cell for the kinetic measurements. The experimental details and additional data are provided as Supplementary Methods.

**Imaging**. The spatial changes in density after the thermochemiluminescence reaction in partially reacted LHP crystals were visualized by computed tomography (CT) using an X View CT Scanner, X500 CT with a working voltage of 120 kV and 100 A. The crystals were attached to a molding clay and scanned at a rate of 7.5 frames per second. The images were processed and videos were recorded using X View CT's default software, eFX-view. The low-light microscopic images were acquired with a low-light microscope using a thermoelectric cooled CMOS detector and a setup as described in detail in the Supplementary Methods.

**X-ray diffraction**. The crystal structures of crystallized LHP and of two isolated reaction products, lophine and the decomposition product dibenzoylamidine (N-benzoylbenzamide), were determined using a Bruker APEX DUO diffractometer. Temperature-dependent powder X-ray diffraction data of LHP were collected with an Anton Paar high-temperature chamber HTK 1200N on Panalytical Empyrean system by using Cu radiation.

**Computations**. Density functional theory (DFT; B3LYP/6-31+G(d,p)) calculations were carried out using Gaussian09, GaussView5 and iboView software (the references are provided as Supplementary References). The optimized ground states were verified by frequency calculation. The IBO charges were calculated by using the PBE functional, the def2-TVVP basis set and the univ-JFIT fit basis.

## Data availability

The X-ray crystallographic coordinates for structures reported in this study have been deposited at the Cambridge Crystallographic Data Centre (CCDC), under deposition

numbers 1885196, 1874149, and 1874147. These data can be obtained free of charge from The Cambridge Crystallographic Data Centre via www.ccdc.cam.ac.uk/data_request/cif.

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

## Acknowledgements

This work was financially supported by New York University Abu Dhabi (NYUAD). The research was in part performed by using the Core Technology Platform resources at NYUAD. We thank Dr. Matthew O'Connor, Dr. Rachid Rezgui and Guowei He for the technical support. The computations were carried out on the High Performance Computing resources at NYUAD. The micro-focus IR experiments were performed at the SPring-8 synchrotron beamline BL43IR under proposal number 2018A1235.

## Author contributions

S.S. incepted the study, designed, and performed the synthesis and characterization, designed and assembled the new experimental setups, and wrote the manuscript. D.P.K. performed crystallographic analysis. N.M.L. recorded some of the spectra. P.C. recorded and analyzed the computed tomography (CT) scans. E.A., L.C. and T.M. recorded the infrared spectra using synchrotron radiation. L.L. assisted with the thermal analysis and powder X-ray diffraction measurements. J.W. assisted the experiments. K.M.S. contributed with the spectroscopic analysis. P.N. co-wrote the manuscript and supervised the work.

## Additional information

**Competing interests:** The authors declare no competing interests.

