## [Peer Review File · Nature Communications]

Reviewers' comments:

Reviewer #1 (Remarks to the Author):

The manuscript presented by Schramm and collaborators summarizes their results with thermochemiluminescence of peroxides in the solid state. It appears that this is the first report of such phenomenon for crystals of peroxides, opening the road for further studies in both basic and applied science. The used experimental apparatus for crystal emission imaging and the possibility of modulating the yield of singlet and triplet states, based on solvent polarity, are two notable aspects of this work. I consider that this manuscript is of interest of the diverse readership of Nature Comm., due to its novelty and scientific impact. It should, therefore, be published after some minor corrections, summarized below.

– In the abstract, the authors state that the described system shows “low chemiluminescent quantum yield $[(2.1 \pm 0.1) \times 10^{-5} \text{ %}]$ ”. Regarding absolute value, this QY is equal to $2.1 \times 10^{-7} \text{ E/mol}$, then? The use of percentage in QY values is somewhat misleading, giving the impression that it's higher than it is. This QY is also reported in units of percentage in page 5, second paragraph.

– In the abstract, the authors state “that the thermochemiluminescence is common for other solid peroxides,” but I believe that they should emphasize that this occurs with crystalline solids.

– In the Introduction, the authors say that bioluminescence “occurs in enzymatic scaffold.” I don't believe that this is the best definition. AMPPD decomposition by alkaline phosphatase is not bioluminescence, but chemiluminescence. Also, I think that it is not a common belief that bioluminescence happens only through enzymatic pathways. The authors must elaborate on this issue.

– Second page, first line, 1,2-dioxetanes and 1,2-dioxetanedione are misspelled.

– When lophine is first mentioned in the Introduction, it's appropriate to furnish its IUPAC name, as 2,4,5-triphenylimidazole or 2,4,5-triphenyl-1H-imidazole.

– When the authors state, in the second paragraph of page 5, that the chemiluminescence QY “is close to the yield reported for the reaction in solution,” reference 9 is cited. I don't believe this is the correct reference.

– In page 5, at the end of the second paragraph, the authors comment on the activation parameters values. I want the authors to address the change in activation entropy further. It is positive in the solid, but negative in both solvents. Therefore, the transition state is more disorganized in the solid, while a certain reorganization is required in condensed media, particularly in NMP. I believe this is consistent with the polar nature of the proposed transition state.

– The authors determined the values for enthalpy and entropy of activation through the Eyring equation. Is this correct at all? By definition, Eyring plots should be used for bimolecular processes, which is not the case here. However, these activation parameters can be estimated using the values for E_a and A from the Arrhenius plots. Probably, this would ultimately lead to similar values, but I believe the use of the Eyring equation out of bimolecular reactions is not correct by principle.

– One brief comment. At the end of page 5, the authors state that their “result points out to simultaneous emission from the singlet and triplet excited states of the same species in solution, and the ratio of the two emissions depends upon its microenvironment.” On the following paragraph, they further discuss this result, base on additional data. I'm used to

chemiluminescence systems where one electronic state, singlet or triplet, is produced more efficiently than the other (with orders of magnitude of difference); the authors also mention that this is a common observation. This unique balance over the distribution of excited state generation, modulated by solvation, is the most exciting outcome of the presented manuscript, in my opinion.

– I do not agree that the formation of dibenzoylamidine occurs after lophine is generated from the decomposition of LHP, as mentioned in the second paragraph on page 7. For this to happen by hydrolysis, as mentioned at some point in the text, water would be needed, which is highly unlikely on a solid matrix and at such high temperatures. More likely, I believe that dibenzoylamidine is generated on a parallel reaction, with LHP being converted to the corresponding 1,2-dioxetane after cyclization, which decomposes through O–O and C–C bonds cleavage, generating the amidine. This would happen analogously to what has been reported by White and collaborators, for the lophine chemiluminescence decomposition in basic solution. The formation of dibenzoylamidine from LHP is promoted by the thermochemiluminescence reaction itself. As the lophine anion is being generated, this is surely basic enough to deprotonate LHP, generate a peroxyanion, that easily forms the 1,2-dioxetane ring by cyclization.

Reviewer #2 (Remarks to the Author):

++Note from the Editor: Please see also attachment.++

The manuscript presented by Naumov and coworkers describes thermochemiluminescent peroxide crystals. As explained by the authors, chemiluminescence in the pure solid state was deemed unfeasible due to the very limited ability of molecules for diffusion required for their reaction. Conversely, thermochemiluminescence should not suffer from this limitation, since the light emission is triggered only by heat, therefore the reaction with other chemical reagents is not required.

The thermochemiluminescent emission of light by pure solid compounds was already presented in previous papers. The authors cited the research work of Roda et al. (ref. 7-10) as “thermochemiluminescence accomplished by heating silica nanoparticles doped with 1,2-dioxetanes”. Actually, the TCL compound was deposited on the heating support as solution, but the spot was left to dry providing a solid layer before the TCL measurement. Moreover, other authors in *Tetrahedron* 2012, 68, 6765 (not cited) studied the thermochemiluminescence of a micro-crystalline layer obtained after evaporation of the solvent. The novelty of the paper proposed by Naumov and coworkers resides “only” in the macroscopic size of the TCL organic crystals and not in the TCL emission from solid state products.

The authors state that the findings described in this manuscript “open prospects for application of organic peroxides as multicolor thermally sensitive solid light-emitting materials” and they declare that lophine hydroperoxide (LHP) is stable at room temperature in the solid state. However, LHP was synthesized at -10°C, the solvent was slowly evaporated below 15 °C to avoid decomposition, the product was crystallized in a freezer at -33 °C and it was stored below -20 °C. In the SI, it was reported that “The pure reaction product thermally decomposes at ca. 110–115 °C, but also degrades slowly when stored over prolonged time at room temperature”.

In my opinion, the thermally labile compounds are characterized by a decomposition kinetic depending on the temperature. The authors should investigate and quantitatively describe how long the solid LHP is stable at room temperature (one week, one month, one year...), indicating also the degradation percentage and the nature of the degradation products. These data should be considered aiming to sensing applications based on solid-state thermochemiluminescence.

The authors selected 110-115°C as triggering temperature on the basis of thermogravimetric analysis (decomposition temperature = 116.5 °C). The authors suggest that (pag. 4, variable-

temperature powder X-ray diffraction, Figure 2A) below 100°C no degradation and light emission were observed (the crystal structure was preserved). Therefore, can they confirm that, by heating the crystal at 80°C for several minutes, no trace of light emission was recorded?

Do the authors suggest also that the degradation process slowly occurring at room temperature is a reaction different from the light emission transformation? This point should be clarified.

Pag. 5: The presented study reveals that the species at 530 nm is the main emission in basic solvents, while the species at 430 nm becomes dominant in non-basic solvents. These findings suggest that two different "emitting species" are dominant in basic and non-basic solvents, respectively.

However, despite the fluorescence/phosphorescence investigation reported, in my opinion, it is not fully demonstrated that these two different "emitting species" are singlet and triplet excited states of the same species (lophine anion in non-basic solvents?).

In different solvents the LHP decomposition pathway could be completely different (see for example *Tetrahedron Letters* 2007, 48, 3109), different pathways could operate simultaneously, the concentration and the nature of the emitting species could vary.

Does Figure 3c regard the thermochemiluminescence spectra or the chemiluminescence spectra (as indicated) of LHP in different solvents? Does LHP emit light in basic solvents without heating? In basic solvents, a chemiluminescent reaction (promoted by the solvent) might compete with the thermochemiluminescent process. The two processes could be different from a mechanistic point of view, while providing the same final products (lophine and dibenzoylamidine).

Pag. 6: The authors should explain in more detail why the fluorescence and phosphorescence spectra of lophine as solid were recorded at -196°C. The mp of lophine (2,4,5-Triphenylimidazole) is 274-278 °C.

Moreover, the several differences present in the spectra of neutral, acidic and basic solutions in chlorobenzene and in N-methylpyrrolidinone require a detailed comment. In particular, the acidic fluorescence profile in CB (Fig. 3E).

If I understand well, the authors compare the fluorescence (383 nm) and the phosphorescence (542 nm) of solid lophine at -196°C with the thermochemiluminescence signal recorded by heating solid LHP at 115°C. A comparison might be made but the systems are quite different. However, it is not clear to me how the recorded data can: 1) exclude that the emitting species in thermochemiluminescent process is excited lophine, 2) prove that the emitting species in thermochemiluminescent process is excited lophine anion.

"The phosphorescence of solid lophine is close to the solid-state thermochemiluminescence signal of LHP (530 nm)". Did the authors suggest that the phosphorescence emission of solid lophine derives from lophine anion without a base?

In my opinion, the phosphorescence emission profiles of lophine around 530 nm are not very different in basic and acidic media.

In conclusion, I am not convinced that the two different emitting species are singlet and triplet excited states of lophine anion.

Concerning the proposed mechanism for the LHP solid-state thermochemiluminescence (Figure 4A), I am not convinced that the lophine anion is the emitting species in TCL process because the experimental findings provided by the authors are not sufficient to prove it.

For example:

- 1) Was a homolytic bond cleavage considered (with the formation of radicals and/or radical-ions)?
- 2) The authors should provide a greater clarification as to exclude the possibility of the excited lophine as emitting species.
- 3) The isolation of lophine as major decomposition product does not mean that lophine (or lophine anion) is the emitting species in thermochemiluminescence. In fact, the recorded low quantum yield could derive from a low amount of the true emitting species.
- 4) In many papers concerning LHP, the proposed decomposition pathway and emitting species are

different:

Ref. 13: J. Am. Chem. Soc. 86, 5685 (1964).

The LHP is thermochemiluminescent above 110° and it is chemiluminescent on treatment with base. For LHP, the quantum efficiency for thermochemiluminescence (ca. 3×10^{-7}) has been found to be about a factor of five smaller than that for chemiluminescence. The chemiluminescence may arise from a triplet-to-singlet transition in the anion of lophine. At -196°C lophine phosphoresces strongly at 523 nm in alcoholic potassium hydroxide and weakly in absolute alcohol. It therefore seems possible that the excited anion of lophine is formed from the decomposition of the anion of LHP and is the light emitter in the chemiluminescence reaction. In the case of the weaker thermochemiluminescence, either free lophine or its anion in smaller concentration may be the emitter.

J. Am. Chem. Soc. 86, 5686 (1964).

All of the lophine-derivatives yielded light on reaction with a base and oxygen, and all of the peroxides yielded light on reaction with base alone. The anion of the hydroperoxide is probably a common intermediate in the two reactions. We found further that salts of diarylarylamidines (III) are products of both reactions and that in the two most efficient cases the fluorescence emissions of the amidine salts match the chemiluminescence emissions of the corresponding lophines. This suggests that these amidine salts are the light emitters in the chemiluminescence. Hydrogen bonding could facilitate the addition leading to the four-membered ring. Light emission occurs from the excited singlet state of compound III and possibly the singlet state is formed directly; on the other hand, the triplet states of V and III may be intermediates as suggested for the chemiluminescence of the phthalic hydrazides.

Ref. 24: Tetrahedron Lett. 48, 3109-3113 (2007).

Lophine peroxide (1a) underwent three different but simultaneous reactions upon treatment with base or heat to provide the corresponding amidine (3a) accompanied with chemiluminescence, imidazole (2a) with singlet oxygen, and a trace of imidazolone (4a).

Ref. 25: Luminescence 22, 72-76 (2007).

Ref. 27: Eur. J. Org. Chem. 2014, 1212-1219 (2014).

A curiosity: were bis(adamantly)-1,2-dioxetane, benzoyl peroxide and rubrene endoperoxide heated as solids or as macroscopic crystals?

Supporting Information:

Chapter 1.3:

The protocol applied to isolate the decomposition products (lophine hydroperoxide dissolved in 50 mL of boiling toluene and stirred under reflux for more than 30 min) is significantly different from the protocol employed to record light emission by thermochemiluminescence (heating a macroscopic crystal). The decomposition products distribution could be different. If the authors work with 1 cm crystals, they could characterize the degradation products directly on the crystal after the light emission.

Chapter 1.7:

The activation parameters were derived from several isothermal chemiluminescence kinetics which were analyzed with Arrhenius and Eyring equations. Were the several isothermal chemiluminescence kinetics carried out at different temperatures? The method and the results should be illustrated in SI.

Movie S7 (SEM micrographs of heated LHP crystals from room temperature to 160 °C) is missing.

In conclusion, my final recommendation is that the proposed work is not fully suitable for the publication in the excellent IF journal Nature Communications.

The paper is well written and presented, the thermochemiluminescence could offer interesting and innovative opportunities (such as applications based on solid-state thermochemiluminescence), the authors presented a multidisciplinary study accomplished employing several modern techniques.

However, many concepts and findings were already known (see for example J. Am. Chem. Soc. 1964, 86, 5685 concerning TCL of LHP).

This is the first example of macroscopic sized TCL organic crystals, but TCL emission from solid state organic products was already reported.

Sensing applications based on thermochemiluminescence of macroscopic organic crystals are very appealing but about this the presented work is in a preliminary stage.

In my opinion, the experimental findings provided by the authors are not sufficient to demonstrate the TCL reaction mechanism and the nature of the emitting species.

Reviewer #3 (Remarks to the Author):

++ Note from the Editor: Please see also attachment. ++

****Lophine****

When refining this structure straight after extracting it from the embedded CIF file, the R factor will be 24.4% -- much higher than the reported 12.19%. I do not understand how this is possible.

Diffraction of this material is reported to 0.83Å -- but in reality, the I/σ drops below the 3-sigma line at around 1Å -- after that, the reflections are mostly noise: we are dealing with a low-resolution structure.

The R_{int} is also very high, and it turns out that his material is twinned. Taking this twinning into account, the R factor will drop to around 11% (using all data).

Cutting the data to exclude noise (OMIT -3 105) will lead to a (just) acceptable structure with an R_1 of 8.54%.

I attach my current refinement, which you may use as a starting point for your new refinement of this structure.

There are no issues with the other two structures and I recommend publication of those.

Response to the comments from the reviewers

We thank all reviewers for the valuable comments which have contributed significantly to improve the quality of the manuscript. We considered all comments, and we tried to address them to the best of our ability. For convenience, in what follows, the original comments from the reviewers are highlighted in blue color, our response is provided in black color, and the text that was modified or added to the manuscript is marked with red color.

Response to the comments from Reviewer #1

Comment: *The manuscript presented by Schramm and collaborators summarizes their results with thermochemiluminescence of peroxides in the solid state. It appears that this is the first report of such phenomenon for crystals of peroxides, opening the road for further studies in both basic and applied science. The used experimental apparatus for crystal emission imaging and the possibility of modulating the yield of singlet and triplet states, based on solvent polarity, are two notable aspects of this work. I consider that this manuscript is of interest of the diverse readership of Nature Comm., due to its novelty and scientific impact. It should, therefore, be published after some minor corrections, summarized below.*

In the abstract, the authors state that the described system shows “low chemiluminescent quantum yield $[(2.1 \pm 0.1) \times 10^{-5} \text{ \%}]$ ”. Regarding absolute value, this QY is equal to $2.1 \times 10^{-7} \text{ E/mol}$, then? The use of percentage in QY values is somewhat misleading, giving the impression that it's higher than it is. This QY is also reported in units of percentage in page 5, second paragraph.

Response to the comment: We thank the reviewer for the generally positive assessment of the contents of the manuscript, and for the constructive suggestions. All suggestions were considered and taken into account in the revised version.

We agree with the reviewer’s remark on the units used to express the quantum yield. In the revised version of the main text and the supplementary material we changed the units of the quantum yield from percent to E/mol:

In the main text:

“...blue-green light with maximum at 530 nm with low chemiluminescent quantum yield $[(2.1 \pm 0.1) \times 10^{-7} \text{ E mol}^{-1}]$.”

“...determined by using absolute calibration methods, was $(2.1 \pm 0.1) \times 10^{-7} \text{ E mol}^{-1}$ ($n = 10$),”

In the Supplementary Material:

“This resulted in a chemiluminescence quantum yield of $2.13 \times 10^{-7} \pm 1.07 \times 10^{-8} \text{ E mol}^{-1}$.”

Comment: *In the abstract, the authors state “that the thermochemiluminescence is common for other solid peroxides,” but I believe that they should emphasize that this occurs with crystalline solids.*

Response to the comment: We thank the reviewer for this suggestion. In the revised version we modified the abstract accordingly:

“With selected 1,2-dioxetane, endoperoxide and aroyl peroxide we also establish that the thermochemiluminescence is common for other crystalline peroxides, and their spectrum is characteristic for the respective decomposition products with the color of the emitted light varying from blue to green to red.”

Comment: *In the Introduction, the authors say that bioluminescence "occurs in enzymatic scaffold." I don't believe that this is the best definition. AMPPD decomposition by alkaline phosphatase is not bioluminescence, but chemiluminescence. Also, I think that it is not a common belief that bioluminescence happens only through enzymatic pathways. The authors must elaborate on this issue.*

Response to the comment: The reviewer has brought up an important point, because there are cases where the light can be generated by using enzymes and this process does not happen in a living organism. We tried to formulate the statement to be consistent with the definition in a recent comprehensive review on bioluminescence and chemiluminescence (Vacher et al., Chem. Rev. (2018) 118, 15, 6927-6974). To define bioluminescence more precisely, in the revised version the paragraph:

*“When this conversion of chemical energy stored within the chemical bonds into visible light occurs in enzymatic scaffold, the process is known as bioluminescence, and is used by tens of thousands of lower biological organisms to communicate, attract prey, or mate.”*¹

was changed to: *“When this light is generated by a living organism, in many cases by catalysis in an enzymatic scaffold, the process is known as bioluminescence, and is used by lower biological organisms to communicate, attract prey, or mate.”*¹

Comment: *Second page, first line, 1,2-dioxetanes and 1,2-dioxetanedione are misspelled.*

Response to the comment: The typo was corrected: *“The high-energy reaction intermediates are usually thermally labile hydroperoxides, 1,2-dioxetanes, 1,2-dioxetaneones or 1,2-dioxetanedione.”*²⁻⁴

Comment: *When lophine is first mentioned in the Introduction, it's appropriate to furnish its IUPAC name, as 2,4,5-triphenylimidazole or 2,4,5-triphenyl-1H-imidazole.*

Response to the comment: In the revised manuscript, we included the IUPAC name of lophine: *“In search for a material that would display thermochemiluminescence in crystalline state, we turned our attention to lophine (2,4,5-triphenyl-1H-imidazole),...”*. We

have also slightly modified this sentence and included the names of Kimura, White, Sonnenberg, Hayashi, and others who with their seminal research work have contributed to the rich photochemistry of lophine and lophine peroxide and their derivatives. The reference to some of their most useful publications are provided as references 12 – 29 (due to the restrictions with the number of citations, we could not provide all publications that are relevant to this topic).

Comment: *When the authors state, in the second paragraph of page 5, that the chemiluminescence QY "is close to the yield reported for the reaction in solution," reference 9 is cited. I don't believe this is the correct reference.*

Response to the comment: We thank the reviewer for catching up this unintentional error. The referencing was accidentally mistaken, and in the revised version it is corrected: *"The chemiluminescence quantum yield, determined by using absolute calibration methods, was $(2.1 \pm 0.1) \times 10^{-7} E \text{ mol}^{-1}$ ($n = 10$), and thus it is close to the yield reported for the reaction in solution.¹³"*

Ref. 13: Sonnenberg, J. & White, D. M. Chemiluminescent and thermochemiluminescent lophine hydroperoxide. *J. Am. Chem. Soc.* 1964, 86, 5685-5686.

Comment: *In page 5, at the end of the second paragraph, the authors comment on the activation parameters values. I want the authors to address the change in activation entropy further. It is positive in the solid, but negative in both solvents. Therefore, the transition state is more disorganized in the solid, while a certain reorganization is required in condensed media, particularly in NMP. I believe this is consistent with the polar nature of the proposed transition state.*

Response to the comment: To clarify this, the following paragraph was added: *"The higher activation parameters in the solid state indicate that the lattice of the crystal increases the energy required for decomposition, while the stabilization by polar solvents points out to a highly polar transition state. Moreover, the activation entropy in the solid is positive, while it is negative in both solvents. Therefore, the transition state in the solid is less ordered in the crystal, while a certain reorganization occurs in solvents, particularly in NMP. This observation is consistent with the polar nature of the proposed transition state (Figure 4A)."*

Comment: *The authors determined the values for enthalpy and entropy of activation through the Eyring equation. Is this correct at all? By definition, Eyring plots should be used for bimolecular processes, which is not the case here. However, these activation parameters can be estimated using the values for E_a and A from the Arrhenius plots. Probably, this would ultimately lead to similar values, but I believe the use of the Eyring equation out of bimolecular reactions is not correct by principle.*

Response to the comment: We thank the reviewer for this comment, which we considered very carefully and in view of the existing literature on bioluminescence. The basic difference between the Arrhenius and Eyring equation is that the first one is empirical, while the latter is based on statistical mechanics principles; both consider that the reaction occurs by formation of an activated complex (transition state). The Eyring equation has been used in unimolecular processes, such as the kinetics of molecular rotors (for a recent example, see *ChemPhysChem*, 2016, 17, 1819-1822). We note that the Eyring equation has also been commonly used in chemiluminescence research to assess the activation parameters (*Chem. Rev.* 2018, 118, 6927-6974, section A.1.7 “Determination of activation parameters and quantum yields”). Therefore, we also applied the equation here to derive values that can be compared to values for other peroxides available from the literature. We agree with the reviewer that both E_a and A can also be derived from an Arrhenius plot. These quantities were thus also derived, and are displayed in Figure 3B. A more detailed explanation on the used experimental method is available from the Supplementary materials, in the section “1.7. Optical spectroscopy”.

Comment: *One brief comment. At the end of page 5, the authors state that their "result points out to simultaneous emission from the singlet and triplet excited states of the same species in solution, and the ratio of the two emissions depends upon its microenvironment." On the following paragraph, they further discuss this result, base on additional data. I'm used to chemiluminescence systems where one electronic state, singlet or triplet, is produced more efficiently than the other (with orders of magnitude of difference); the authors also mention that this is a common observation. This unique balance over the distribution of excited state generation, modulated by solvation, is the most exciting outcome of the presented manuscript, in my opinion.*

Response to the comment: We share the reviewer’s excitement over this result, and we were very surprised to find out that we were able to record emission from triplet and singlet emission state, and that the ratio of the two strongly depends on the microenvironment in which the reaction takes place. We reiterate that such a system could be important as a ratiometric dye in order to directly asses these microenvironment-dependent properties that directly affect its emission.

Comment: *I do not agree that the formation of dibenzoylamidine occurs after lophine is generated from the decomposition of LHP, as mentioned in the second paragraph on page 7. For this to happen by hydrolysis, as mentioned at some point in the text, water would be needed, which is highly unlikely on a solid matrix and at such high temperatures. More likely, I believe that dibenzoylamidine is generated on a parallel reaction, with LHP being converted to the corresponding 1,2-dioxetane after cyclization, which decomposes through O–O and C–C bonds cleavage, generating the amidine. This would happen analogously to what has been reported by White and collaborators, for the lophine chemiluminescence decomposition in basic solution. The formation of dibenzoylamidine from LHP is promoted by the thermochemiluminescence reaction itself. As the lophine anion is being generated, this is surely basic enough to deprotonate LHP, generate a*

peroxyanion, that easily forms the 1,2-dioxetane ring by cyclization.

Response to the comment: We thank the reviewer for the detail assessment of the reaction mechanism, which is the central part of this work. We do share with the reviewer their opinion that the formation of dibenzoylamidine does not occur after lophine is generated by decomposition of LHP. The water that would be needed for this reaction is unlikely to be found in the solid state at such high temperature. We also agree that a purported side reaction leads to the formation of a dioxetane which results in formation of dibenzoylamidine after its decomposition. We also agree that the lophine anion should be basic enough to deprotonate LHP and to generate a peroxyanion which forms the 1,2-dioxetane ring by cyclization. In the revised version, we have added this hypothesis that could also explain the formation of the side-product:

“We hypothesize that the side-product (dibenzoylamidine) might be formed by a parallel reaction. In that scenario, initiated by the basic lophine anion that is formed in the main thermochemiluminescence reaction, LHP is converted to the corresponding 1,2-dioxetane after cyclization, and the product decomposes through O–O and C–C bonds cleavage to generate dibenzoylamidine. This hypothesis is consistent with the reported base-induced decomposition of LHP reported by White et al.^{13,14}”

Figure 4 was modified to reflect these reaction pathways:

Response to the comments from Reviewer #2

Comment: *The manuscript presented by Naumov and coworkers describes thermochemiluminescent peroxide crystals. As explained by the authors, chemiluminescence in the pure solid state was deemed unfeasible due to the very limited ability of molecules for diffusion required for their reaction. Conversely, thermochemiluminescence should not suffer from this limitation, since the light emission is triggered only by heat, therefore the reaction with other chemical reagents is not required. The thermochemiluminescent emission of light by pure solid compounds was already presented in previous papers.*

Response to the comment: We would like to thank the reviewer—who, based on the very detailed comments, is apparently a knowledgeable expert in this research field—for the extremely careful reading, very detailed comments, and balanced assessment of the work described in this manuscript. The corresponding author of this manuscript has about two decades of experience with publishing, and throughout the years has learned to appreciate a thorough and rigorous assessment of a scientific report such as this one, which can only contribute to improvement of the quality of a research contribution. The authors also appreciate the effort and time that the reviewer has apparently taken to assess the research results, particularly in view of the reaction mechanism within the context of the existing literature on (thermo)chemiluminescence.

We also do share the reviewer's sentiment that conceptually the work presented here might not appear very novel in view of the current interest in emission from nano-sized crystals. The authors of this work are aware of both earlier articles, which mainly focus on the preparation and solution-state chemistry of peroxides, largely contributed by the research groups of Adam and Baader, as well as of important contributions related to application aspects of the TCL, a research led by Roda, Kricka, Gilbert and other authors, and especially the seminal works on lophine photochemistry by Kimura, White, Sonnenberg and others, to this research field, particularly in view of the limited amount of data on the TCL from crystals. We would like to reiterate that the main goal of this work was to build on the previous work in this research field by detecting emission of light from peroxide crystals of macroscopic size. For that purpose, in a course of research work that extended over more than two years, we have built a special experimental setup (a low-light microscope) and a couple of other devices, and we used a number of contemporary experimental techniques to be able to detect the light emission from centimeter-size peroxide crystals and to unravel the reaction mechanism. We hope that these results will contribute to further understanding of the mechanism of thermochemiluminescence, particularly in the solid state, where the ordered nature of the molecules in the crystal lattice can sometimes result in different outcome of the underlying photophysical and photochemical processes.

Comment: *The authors cited the research work of Roda et al. (ref. 7-10) as “thermochemiluminescence accomplished by heating silica nanoparticles doped with 1,2-dioxetanes”. Actually, the TCL compound was deposited on the heating support as solution, but the spot was left to dry providing a solid layer before the TCL measurement. Moreover, other authors in Tetrahedron 2012, 68, 6765 (not cited) studied the*

thermochemiluminescence of a micro-crystalline layer obtained after evaporation of the solvent. The novelty of the paper proposed by Naumov and coworkers resides “only” in the macroscopic size of the TCL organic crystals and not in the TCL emission from solid state products.

Response to the comment: We thank the reviewer for bringing to our attention the article of Gilbert and the collaborators that beautifully illustrates application of thermochemiluminescence. In the revised version of the manuscript, we have included the Tetrahedron article as a new reference (now reference #10; due to constraints with the number of references, we had to replace one of the existing references with this new reference). The intent with the work presented in our manuscript was to build on the existing results on TCL nanocrystals (Roda and the collaborators) and microcrystalline powder (now included with the article of Gilbert and collaborators) by adding to this research results obtained on macroscopically large crystals (mm to cm in size). We hope that these results will be seen as a valuable contribution to this growing research field, and we also hope that they will bring more interest in both fundamental and applied aspects of thermochemiluminescence, particularly by providing a platform to study the photophysical aspects of thermochemiluminescence in an ordered, long-range scale.

Comment: *The authors state that the findings described in this manuscript “open prospects for application of organic peroxides as multicolor thermally sensitive solid light-emitting materials” and they declare that lophine hydroperoxide (LHP) is stable at room temperature in the solid state. However, LHP was synthesized at -10°C, the solvent was slowly evaporated below 15 °C to avoid decomposition, the product was crystallized in a freezer at -33 °C and it was stored below -20 °C. In the SI, it was reported that “The pure reaction product thermally decomposes at ca. 110–115 °C, but also degrades slowly when stored over prolonged time at room temperature”. In my opinion, the thermally labile compounds are characterized by a decomposition kinetic depending on the temperature. The authors should investigate and quantitatively describe how long the solid LHP is stable at room temperature (one week, one month, one year...), indicating also the degradation percentage and the nature of the degradation products. These data should be considered aiming to sensing applications based on solid-state thermochemiluminescence.*

Response to the comment: We thank the referee for pointing out the necessity for further assessment of the thermal stability of the lophine hydroperoxide. This certainly is a very important aspect in the light of future applications of this material. Intrigued by the reviewer’s comment, we performed a detailed study of its stability. To that end, we measured the isothermal decomposition rate of the compound at elevated temperature, in 10 °C-intervals between 50 °C and 80 °C, by using thermogravimetry. We note that no traces of decomposition could be observed below 50°C even after 48 h, so we could assume that the compound is stable under those conditions. From the decomposition rates determined at different temperatures, we extrapolated the decomposition rate for room temperature by using the van 't Hoff’s rule:

$$Q_{10} = \left(\frac{k_2}{k_1}\right)^{\frac{10\text{ K}}{T_2 - T_1}}$$

with Q_{10} for the factor of which the reaction time, that is, the decomposition time, increases with a 10 K-increment. The results are shown below:

Temperature / °C	$k_{\text{obs}} / \text{min}^{-1}$	Q_{10}
50	$2.49 \times 10^{-6} \pm 1.06 \times 10^{-9}$	7.28
60	$8.80 \times 10^{-6} \pm 8.28 \times 10^{-9}$	2.25
70	$1.41 \times 10^{-5} \pm 1.86 \times 10^{-8}$	4.40
80	$6.92 \times 10^{-5} \pm 8.21 \times 10^{-8}$	

From this data it was possible to calculate the average Q_{10} , $Q_{\text{ave}} = 4.28 \pm 2.29$. Using $k_1 = k_{25^\circ\text{C}}$, $k_2 = k_{50^\circ\text{C}}$, $T_2 = 50^\circ\text{C}$ and $T_1 = 25^\circ\text{C}$, the average decomposition rate at room temperature was estimated to $k_{25^\circ\text{C}} = 4.86 \times 10^{-8} \pm 2.31 \times 10^{-8} \text{ min}^{-1}$. This rate equals a half-life time of $t_{1/2} = 27.2 \pm 12.9$ years. Therefore, we conclude that LHP is rather stable at room temperature and could be considered for practical applications. A section explaining these experiments and the calculations was included in the revised version of the supplementary information and the text that details the preparation procedures was revised accordingly.

Changes made in the Supplementary material:

“To assess the stability of LHP at room temperature, the isothermal rate of decomposition was also determined in 10 °C (10 K)-intervals between 50 °C and 80 °C. No detectable decomposition could be detected below 50 °C even after 48 hours, the longest time attainable with our experimental setup. Instead, the decomposition rate at room temperature was determined by extrapolation of the decomposition rates measured at elevated temperatures by using the van't Hoff's rule:

$$Q_{10} = \left(\frac{k_2}{k_1}\right)^{\frac{10\text{ K}}{T_2 - T_1}}$$

with Q_{10} as the factor whose reaction (decomposition) time, increases with a 10 °C-increment. The values in Supplementary Table 6 were obtained.

Supplementary Table 6. Rate constants and Q values extracted from the isothermal kinetic measurements

Temperature / °C	$k_{\text{obs}} / \text{min}^{-1}$	Q_{10}
50	$2.49 \times 10^{-6} \pm 1.06 \times 10^{-9}$	7.28

60	$8.80 \times 10^{-6} \pm 8.28 \times 10^{-9}$	2.25
70	$1.41 \times 10^{-5} \pm 1.86 \times 10^{-8}$	4.40
80	$6.92 \times 10^{-5} \pm 8.21 \times 10^{-8}$	

From this data, the average Q_{10} was calculated, $Q_{ave} = 4.28 \pm 2.29$. With $k_1 = k_{25\text{ }^\circ\text{C}}$, $k_2 = k_{50\text{ }^\circ\text{C}}$, $T_2 = 50\text{ }^\circ\text{C}$ and $T_1 = 25\text{ }^\circ\text{C}$, the average decomposition rate at room temperature was estimated to $k_{25\text{ }^\circ\text{C}} = 4.86 \times 10^{-8} \pm 2.31 \times 10^{-8}\text{ min}^{-1}$. This rate constant corresponds to half-lifetime of $t_{1/2} = 27.2 \pm 12.9$ years. This lifetime indicates that LHP is sufficiently thermally stable to be considered for practical applications in the future.”

Changes made in the preparation procedure for LHP in the Supplementary Material:

“The solvent was slowly evaporated. TGA studies (see Section 1.11) indicated that LHP is stable at room temperature with years, and significant decomposition can only be detected over $50\text{ }^\circ\text{C}$. Thus, it is recommend that all workup procedures are performed below $50\text{ }^\circ\text{C}$. Recrystallization of the crude product was attempted from various solvents (CS_2 , ethyl acetate, THF, DMF, dioxane, DCM). Best results were obtained by crystallization in a freezer ($-33\text{ }^\circ\text{C}$) from ethanol as colorless large (0.1–1 cm) block-shaped crystals. Yield: 78% (after recrystallization). The pure reaction product thermally decomposes at ca. $110\text{--}115\text{ }^\circ\text{C}$.”

Comment: *The authors selected $110\text{--}115^\circ\text{C}$ as triggering temperature on the basis of thermogravimetric analysis (decomposition temperature = $116.5\text{ }^\circ\text{C}$). The authors suggest that (pag. 4, variable-temperature powder X-ray diffraction, Figure 2A) below 100°C no degradation and light emission were observed (the crystal structure was preserved). Therefore, can they confirm that, by heating the crystal at 80°C for several minutes, no trace of light emission was recorded?*

Response to the comment: We confirm that heating of a crystalline sample of LHP for several minutes at $80\text{ }^\circ\text{C}$ does not result in emission of light. Moreover, there is no evolution of gas at 80°C that confirms there is no breaking of C-O bonds at this temperature. In support of these conclusions, we performed additional experiments The plot shown below was included in the revised version of the supplementary material. It shows a typical kinetics trace of the thermochemiluminescence reaction of LHP crystal recorded at $130\text{ }^\circ\text{C}$ and at $80\text{ }^\circ\text{C}$. As it can be concluded from the plot, there is not emission at $80\text{ }^\circ\text{C}$.

Supplementary Figure 14. Kinetic traces of the thermochemiluminescence reaction of LHP crystals at 130 °C and 80 °C.”

This plot was included in the supplementary material together with a short comment:

“We note that in order to observe light emission, the LHP crystals have to be heated to their decomposition temperature (116.5 °C). Heating below this temperature does not result in any detectable emission of light (Supplementary Figure 14).”

Comment: Do the authors suggest also that the degradation process slowly occurring at room temperature is a reaction different from the light emission transformation? This point should be clarified.

Response to the comment: We thank the referee for bringing up a very important question. Initially, and based on the results on most of the other peroxides which are known to be notoriously unstable, we did expect very slow decomposition of LHP at room temperature. However, we were not able to detect any light emission at room temperature and to observe gas formation, which indicated that the LHP is stable. Further confirmation of its stability was provided by the additional TGA experiments, explained in the response to the previous comment. These experiments confirmed that there is no significant decomposition below 50 °C. Nevertheless, as a measure of safety precaution, we recommended storage at low temperature (-20 °C) as this is common practice for storage of peroxides. We believe that this warning should remain part of the experimental part.

To reflect this, the main text was modified by including the TGA results, as described above, as well as the following sentence:

“Although thermogravimetric analysis indicated that there is no significant decomposition at room temperature, for safety reasons the crystals were stored below $-20\text{ }^{\circ}\text{C}$.”

Comment: *Pag. 5: The presented study reveals that the species at 530 nm is the main emission in basic solvents, while the species at 430 nm becomes dominant in non-basic solvents. These findings suggest that two different “emitting species” are dominant in basic and non-basic solvents, respectively. However, despite the fluorescence/-phosphorescence investigation reported, in my opinion, it is not fully demonstrated that these two different “emitting species” are singlet and triplet excited states of the same species (lophine anion in non-basic solvents?). In different solvents the LHP decomposition pathway could be completely different (see for example Tetrahedron Letters 2007, 48, 3109), different pathways could operate simultaneously, the concentration and the nature of the emitting species could vary.*

Response to the comment: We appreciate the reviewer’s concerns, and in the meantime we tried to address them to the best of our ability. Figure 3A clearly shows that there is only one maximum in the thermochemiluminescence emission from the solid-state reaction and that this maximum does not change over time. Moreover, the data shows clear first order kinetics for the solid-state thermochemiluminescence, and this is in agreement with the hypothesis of only one high-energy intermediate in the solid-state thermochemiluminescence reaction that results in formation of one emitting molecule, rather than multiple reactions that result in light emission. It can be clearly seen that besides the maximum at 530 nm the solid-state thermochemiluminescence spectrum (Figure 1A) shows a shoulder around 430 nm. Since all our experimental data points out that there is only one chemical species that emits light, we have closely studied the photoluminescence of the proposed emitting molecule, lophine, under different protonation conditions and in different phases (Figure 3, panels D-F). The results in basic media are particularly helpful to explain the emission at 530 nm, which results from the triplet emission of the lophine anion, and the peak at 430 nm, which results from the singlet emission of the lophine anion. We believe that perhaps our explanation in the original text was not sufficiently clear, and could have led to confusion. Therefore, in the revised version of the manuscript we have revised the discussion in the main text. We also included an additional figure in the supporting information which shows that the kinetics for the solid-state thermochemiluminescence is clearly first order and thus it points out to only one reaction pathway of one high-energy intermediate which results in light emission.

The following section in the main text was modified:

“Based on the phosphorescence emission of lophine in basic solid matrices (530 nm) and the thermochemiluminescence of LHP in basic solvents and in the solid state (530 nm), together with the result that the maxima of the thermochemiluminescence reaction in the solid state does not change over time (Figure 3A) and that its kinetics are of first order

(Supplementary Figure 15), we hypothesize that the actual emitter in the solid-state thermochemiluminescence of LHP is the excited triplet state of deprotonated lophine.”

The following text was added to section 1.7 in the Supplementary Material:

“The kinetic trace follows a first order kinetic model ($R^2 > 0.99$). The kinetics constants were derived from this model (Supplementary Figure 15).”

Supplementary Figure 15. Kinetic traces of the thermochemiluminescence reaction of LHP crystals at 130 °C (black) and the fit with a first order kinetic model (red)

Comment: Does Figure 3c regard the thermochemiluminescence spectra or the chemiluminescence spectra (as indicated) of LHP in different solvents?

Response to the comment: Figure 3c shows thermochemiluminescence spectra of LHP on various solvents. We apologize for the imprecise usage of the term ‘chemiluminescence’, which is now replaced with ‘thermochemiluminescence’.

“(C) Chemiluminescence spectra of LHP in 12 solvents ranging from...”

was changed to:

“(C) Thermochemiluminescence spectra of LHP in 12 solvents ranging from...”

Comment: Does LHP emit light in basic solvents without heating?

Response to the comment: In none of the experiments that we performed and in none of the tested basic solvents LHP showed chemiluminescence by itself, without heating. In order to provide further experimental support to this conclusion, we prepared a solution of LHP in the basic solvent NMP and recorded (under identical conditions) the light emission spectrum at room temperature and at 110 °C. The results are shown below and confirm that there is no light emission in pure NMP at room temperature.

Comment: *In basic solvents, a chemiluminescent reaction (promoted by the solvent) might compete with the thermochemiluminescent process. The two processes could be different from a mechanistic point of view, while providing the same final products (lophine and dibenzoylamidine).*

Response to the comment: We agree with the reviewer that strongly basic solvent could indeed induce chemiluminescence reaction of lophine hydroperoxide. In order to exclude this process under the experimental conditions that we have selected for our experiments, we have performed reference experiments in basic solvent and monitored the reaction by NMR at room temperature and 110 °C, as stated above. These experiments did not show light emission under the given experimental conditions in pure NMP without heating.

Comment: *Pag. 6: The authors should explain in more detail why the fluorescence and phosphorescence spectra of lophine as solid were recorded at -196 °C. The mp of lophine (2,4,5-Triphenylimidazole) is 274-278 °C.*

Response to the comment: We would like to confirm that the photoluminescence spectra of lophine were recorded at -196°C in order to extend the lifetime of the excited triplet state so as to be able to record a time-gated phosphorescence spectrum. In the revised version, this detail was explained in the spectroscopy section in the Supplementary Material:

“The photoluminescence spectra of lophine were recorded at -196°C in order to extend the lifetime of the excited triplet state and to be able to record a time-gated phosphorescence spectrum.”

Comment: *Moreover, the several differences present in the spectra of neutral, acidic and basic solutions in chlorobenzene and in N-methylpyrrolidinone require a detailed comment. In particular, the acidic fluorescence profile in CB (Fig. 3E).*

Response to the comment: We thank the reviewer for pointing out that our explanation of the spectra in neutral, acidic and basic media are not sufficiently detailed. In order to clarify the discussion, we added a more detailed description of these spectra to the main text:

“The emission spectra recorded in neutral conditions closely resemble the solid-state spectra. Contrary to this, the spectra recorded in acidic conditions are very different between CB and NMP. While the spectrum in CB shows a complex fluorescence emission with multiple maxima between 450 and 510 nm, the fluorescence maximum in acidic NMP lies very close to the one in neutral medium (ca. 380 nm). The results in basic media are especially striking, and in both solvents lophine shows fluorescence emission around 430 nm and phosphorescence emission around 530 nm.”

Comment: *If I understand well, the authors compare the fluorescence (383 nm) and the phosphorescence (542 nm) of solid lophine at -196°C with the thermochemiluminescence signal recorded by heating solid LHP at 115°C . A comparison might be made but the systems are quite different. However, it is not clear to me how the recorded data can: 1) exclude that the emitting species in thermochemiluminescent process is excited lophine, 2) prove that the emitting species in thermochemiluminescent process is excited lophine anion. “The phosphorescence of solid lophine is close to the solid-state thermochemiluminescence signal of LHP (530 nm)”. Did the authors suggest that the phosphorescence emission of solid lophine derives from lophine anion without a base?*

Response to the comment: We do not suggest that the phosphorescence emission of solid lophine derives from lophine anion without a base. However, we provide evidence that the emitter of the solid-state thermochemiluminescence reaction is the anion of lophine. The formation of such an anion seems to be feasible given the suggested mechanism delineated in Figure 4, which was inferred based on the collection of experimental data. Indeed, to provide further support, we investigated the spectroscopic

properties of the lophine anion by deprotonating the main product of the chemiluminescence reaction (lophine) and we recorded its fluorescence and phosphorescence spectra. While the fluorescence spectra (which reflect an emission from the excited singlet state) reflect very well the thermochemiluminescence spectra recorded in non-polar solvents, the phosphorescence spectrum (which corresponds to emission from the excited triplet state) of deprotonated lophine reflects very well the thermochemiluminescence spectra recorded in polar basic solvents and in the solid state. Based on this evidence, we conclude that the solid-state thermochemiluminescence is a result of triplet emission of deprotonated lophine.

Comment: *Concerning the proposed mechanism for the LHP solid-state thermochemiluminescence (Figure 4A), I am not convinced that the lophine anion is the emitting species in TCL process because the experimental findings provided by the authors are not sufficient to prove it. For example: 1) Was a homolytic bond cleavage considered (with the formation of radicals and/or radical-ions)?*

Response to the comment: We share reviewer's concerns, and we did indeed consider the possibility of a homolytic bond cleavage during this work. Such event would result in the formation of a lophine radical. The lophine radicals have been studied extensively in the past in regards to their photochromic, piezochromic and thermochromic properties. Some notable references related to these species, for example, are: D. White, J. Sonnenberg, *J. Am. Chem. Soc.* 1966, 88, 3825-3829; M. Koko, H. Taro, *Bull. Chem. Soc. Jpn.* 1970, 43, 429-438. The lophine radicals have been described as rather stable and strongly "reddish-purple" colored species (for example, see M. Koko, H. Taro, *Bull. Chem. Soc. Jpn.* 1970, 43, 429-438). In none of our experiments we have encountered any indication of such intense "reddish-purple" coloration, neither we have detected any of these lophine radicals or their dimers in our UHPLC-MS experiments. Based on this, we conclude that such species are not formed under the experimental conditions used in this study and we therefore exclude a homolytic bond cleavage.

In order to clarify this point, in the revised version of the manuscript we added a short discussion to the main text:

"We note that during the experiments we did not encounter any indication of intense reddish-purple coloration that is common for the lophine radical and would indicate homolytic bond cleavage, neither did we observe any lophine radicals or their dimers in the UHPLC-MS experiments (Supplementary Figure 3). Therefore, we conclude that radicals are not formed under our experimental conditions and we exclude the possibility for a homolytic bond cleavage."

Comment: *2) The authors should provide a greater clarification as to exclude the possibility of the excited lophine as emitting species.*

Response to the comment: In the revised version of the main text, we have added a discussion which clarifies that the photophysical properties of lophine alone can not explain the experimental solid-state thermochemiluminescence emission spectra, and that these spectral features are very well described by the basic fluorescence and phosphorescence spectra of deprotonated lophine. The following text was added/modified:

“While the phosphorescence emission of lophine is close to the solid-state thermochemiluminescence of LHP (530 nm), its fluorescence is far from the LHP emission in solution (430 nm). Therefore, the photophysical properties of lophine can not explain the observed solid-state thermochemiluminescence emission spectra, and necessitated deeper insight into the lophine photochemistry. Consequently, the photoluminescence spectra in solidified neutral, basic and acidic matrices of chlorobenzene (CB) and N-methylpyrrolidinone (NMP) were recorded at $-196\text{ }^{\circ}\text{C}$ (Figure 3E and 3F).”

“Based on the phosphorescence emission of lophine in basic solid matrices (530 nm) and the thermochemiluminescence of LHP in basic solvents and in the solid state (530 nm), and taking into account that the maxima of the thermochemiluminescence reaction in the solid state does not change over time (Figure 3A) and that its kinetics is of first order (Supplementary Figure 15), we hypothesized that the actual emitter in the solid-state thermochemiluminescence of LHP is the excited triplet state of deprotonated lophine.”

Comment: 3) *The isolation of lophine as major decomposition product does not mean that lophine (or lophine anion) is the emitting species in thermochemiluminescence. In fact, the recorded low quantum yield could derive from a low amount of the true emitting species.*

Response to the comment: We entirely agree with the reviewer that the isolation of lophine as the main decomposition product does not necessarily imply that this molecule also is the emitter in the solid-state thermochemiluminescence. Indeed, we have explained in the manuscript that in fact the collective results point out to the lophine ion rather than the neutral molecule as emitting state in the solid state. Our conclusion is based on spectroscopic results rather than on the product analysis. As we have detailed above, spectroscopically, it is very unlikely that pure lophine is the emitter in this reaction because its photoluminescence spectra are not in good agreement with the observed solid-state thermochemiluminescence emission spectra. Contrary to this, the experimental phosphorescence spectra of deprotonated lophine reflect very well the solid-state thermochemiluminescence emission spectra while the fluorescence emission of deprotonated lophine also fits very well the thermochemiluminescence emission spectra in non-polar solvents. Since emission in the solid state from an excited triplet state necessitates intersystem crossing and such an intersystem crossing is quantum mechanical forbidden, only a very small amount of molecules is expected to cross to their excited triplet state and even smaller number of them will emit from this excited triplet state. This implies very low quantum yield, as it is indeed observed in the experiments.

To clarify this, we attempted to improve the discussion section in the revised manuscript that pertains to this aspect of the results:

“Therefore, the photophysical properties of lophine can not explain the observed solid-state thermochemiluminescence emission spectra, and necessitated deeper insight into the lophine photochemistry. Consequently, the photoluminescence spectra in solidified neutral, basic and acidic matrices of chlorobenzene (CB) and N-methylpyrrolidinone (NMP) were recorded at $-196\text{ }^{\circ}\text{C}$ (Figure 3E and 3F). The emission spectra recorded in neutral conditions closely resemble the solid-state spectra. Contrary to this, the spectra recorded in acidic conditions are very different between CB and NMP. While the spectrum in CB shows a complex fluorescence emission with multiple maxima between 450 and 510 nm, the fluorescence maximum in acidic NMP lies very close to the one in neutral medium (ca. 380 nm). The results in basic media are especially striking, and in both solvents lophine shows fluorescence emission around 430 nm and phosphorescence emission around 530 nm. Based on the phosphorescence emission of lophine in basic solid matrices (530 nm) and the thermochemiluminescence of LHP in basic solvents and in the solid state (530 nm), and taking into account that the maxima of the thermochemiluminescence reaction in the solid state does not change over time (Figure 3A) and that its kinetics is of first order (Supplementary Figure 15), we hypothesized that the actual emitter in the solid-state thermochemiluminescence of LHP is the excited triplet state of deprotonated lophine. The triplet emission necessitates intersystem crossing. Such an intersystem crossing is quantum mechanical forbidden. Only a very small amount of molecules will actually cross to their excited triplet state and even less will emit from this excited triplet state. Hence, the observed thermochemiluminescent quantum yield is inherently very low.”

Comment: *4) In many papers concerning LHP, the proposed decomposition pathway and emitting species are different:*

Ref. 13: J. Am. Chem. Soc. 86, 5685 (1964).

The LHP is thermochemiluminescent above 110° and it is chemiluminescent on treatment with base. For LHP, the quantum efficiency for thermochemiluminescence (ca. 3×10^{-7}) has been found to be about a factor of five smaller than that for chemiluminescence. The chemiluminescence may arise from a triplet-to-singlet transition in the anion of lophine. At -196°C lophine phosphoresces strongly at 523 nm in alcoholic potassium hydroxide and weakly in absolute alcohol. It therefore seems possible that the excited anion of lophine is formed from the decomposition of the anion of LHP and is the light emitter in the chemiluminescence reaction. In the case of the weaker thermochemiluminescence, either free lophine or its anion in smaller concentration may be the emitter.

Response to the comment: We thank the reviewer for the very detailed guidance through the earlier literature on this subject. We are aware of most of these articles, and we are happy to have learned from the reviewer’s comments more about a few additional articles related to this system. We have inspected very carefully all articles in great detail, and we have compared the results presented in our manuscript with those published in

the related articles. While we would like to refrain here from commenting on the consistency of the results in the already published results as that is outside the scope of our study, we have carefully checked the explanation of the mechanism that we provide in our manuscript against the content of these articles, and in continuation we present a brief summary of our response in regards to each individual article that was mentioned in the comment.

The overall results in the first article that the reviewer has highlighted in this comment (*J. Am. Chem. Soc.* 1964, 86, 5685), which describes the thermochemiluminescence in solution but not in the solid state, indeed agree very well with the experimental results and the conclusions that we present in our manuscript. For instance, the authors of the article have concluded that “*The chemiluminescence may arise from a triplet-to-singlet transition in the anion of Lophine...*” and “*...that the excited anion of lophine is formed from the decomposition of the anion of LHP and is the light emitter...*”. We note that our experimental results are consistent and lead to the exactly same conclusion. This article has been cited as reference #13.

Comment: *J. Am. Chem. Soc.* 86, 5686 (1964).

All of the lophine-derivatives yielded light on reaction with a base and oxygen, and all of the peroxides yielded light on reaction with base alone. The anion of the hydroperoxide is probably a common intermediate in the two reactions. We found further that salts of diarylarylamidines (III) are products of both reactions and that in the two most efficient cases the fluorescence emissions of the amidine salts match the chemiluminescence emissions of the corresponding lophines. This suggests that these amidine salts are the light emitters in the chemiluminescence.

Hydrogen bonding could facilitate the addition leading to the four-membered ring. Light emission occurs from the excited singlet state of compound III and possibly the singlet state is formed directly; on the other hand, the triplet states of V and III may be intermediates as suggested for the chemiluminescence of the phthalic hydrazides.

Response to the comment: We thank the reviewer for pointing this article, which elaborates the chemiluminescence reaction of unsubstituted lophine and its derivatives, however it does not include the *thermochemiluminescence* reaction of lophine *hydroperoxide*. We are aware that this interesting base-induced chemiluminescence reaction has been intensively studied by several research groups in the past. In our manuscript, we provided a summary of these earlier studies as references 12-21. At the essence, addition of very strong base to lophine or its derivatives induces formation of dioxetane, which is subsequently cleaved and results in a reaction product, dibenzoylamidine, in excited state. Its relaxation to the ground state gives rise to emission by chemiluminescence. Based on our experimental data described in our manuscript, we suggest that the solid-state thermochemiluminescence reaction is mechanistically different from the solution-state base-induced chemiluminescence reaction of lophine derivatives. This conclusion can also be understood within the context of the quantum yield of light emission. While in case of the thermochemiluminescence reaction of lophine hydroperoxides the yield is on the order of 10^{-7} E mol⁻¹, for the base-induced chemiluminescence reaction of lophine derivatives it is several orders of magnitudes

larger, up to 3.7×10^{-3} , as reported by Kimura and collaborators (M. Kimura, H. Nishikawa, H. Kura, H. Lim, E. H. White, *Chem. Lett.* 1993, 22, 505-508).

Comment: *Ref. 24: Tetrahedron Lett. 48, 3109-3113 (2007).*

Lophine peroxide (1a) underwent three different but simultaneous reactions upon treatment with base or heat to provide the corresponding amidine (3a) accompanied with chemiluminescence, imidazole (2a) with singlet oxygen, and a trace of imidazolone (4a).

Response to the comment: In this study, the authors explored the details of the stereoselective thermal rearrangement of chiral lophine peroxides, in particular, their silyl derivatives, in solution. In addition to oxygen, lophine and dibenzoylamidine were obtained as products, both of which were also observed in our work, the authors also discovered a trace of an imidazolone derivative. We carefully checked our analytical data for traces of such imidazolone derivative however we were not able to find any indication for the formation of the side-product. Nevertheless, in order to clarify this point, in the revised version we have included a section that explicitly mentions the literature where the formation of the imidazolone is described: *“The formation of trace of an imidazolone as side-product has been reported for the stereoselective thermal rearrangement of chiral lophine peroxides, in particular silyl derivatives, in solution,²⁴ however we could not observe this product by thermally induced chemiluminescence in any of our experiments.”*

Comment: *Ref. 25: Luminescence 22, 72-76 (2007).*

Response to the comment: We inspected very carefully this reference which describes that singlet oxygen is formed in the base-induced chemiluminescence reaction of lophine hydroperoxide, and a copy of the experimental section is provided below.

8.35 (d, $J = 7.0$ Hz, 2H) 13.52 (bs); UV-vis λ_{max} (CH_2Cl_2) 303 (log ϵ 1.85) nm, 401 (log ϵ 1.23) nm; MS (FAB) m/z 372 ($\text{M}^+ + 1$); anal. calcd for $\text{C}_{23}\text{H}_{21}\text{N}_3$, C, 74.39; H, 5.72; N, 11.28; found C, 74.37; H, 5.70; N, 11.31.

Reaction of lophine peroxides in the presence of 1,3-DPBF

A solution of **2** and 1,3-DPBF in CH_2Cl_2 was stirred under N_2 protection in a dark room, after which a solution of KOH in methanol (0.5 mol/L) was added. The reaction mixture was stirred for 30 min under N_2

protection, and then aqueous HCl (1 N) was added to neutralize the base. The organic layer was separated, washed and dried with anhydrous MgSO_4 . The solvent was removed under reduced pressure, and then the products were analysed with $^1\text{H-NMR}$ and isolated with column chromatography (silica, hexane and ethyl acetate).

Apparatus

HPLC, HITACHI 655 Liquid Chromatograph, HITACHI 561 Recorder; column, Inertsil ODS-3,

We would like to point out that in our work we studied the solid-state thermally induced chemiluminescence of lophine hydroperoxide in crystalline state. As clearly illustrated in the figure 1L-O of the main manuscript, crystalline lophine hydroperoxide also releases oxygen when thermally decomposed.

Comment: *Ref. 27: Eur. J. Org. Chem. 2014, 1212-1219 (2014).*

Response to the comment: This article discusses the base-induced chemiluminescence reaction of lophine hydroperoxide and its derivatives in solution. Specifically, it focuses on the chemiexcitation efficiency of asymmetrically substituted lophine derivatives that can form different regioisomers during the base-induced decomposition. It is concluded that these different isomers play an important role in the base-induced chemiluminescence reaction due to their different chemiluminescence efficiencies.

We would like to note that our results are not in contradiction with this work because the focus of our study was not the understanding of the base-induced solution chemiluminescence of lophine or its derivatives. Instead, our work focuses on the solid-state thermochemiluminescence of lophine hydroperoxide, a conceptually and mechanistically different process. We discuss in our paper the possible reaction mechanism of the solid-state thermochemiluminescence. Based on the experimental observation that the solid-state thermochemiluminescence reaction goes in hand with the release of oxygen, we suggest that—in contrast to the base-induced chemiluminescence reaction in solution—no dioxetane is formed as the main high-energy intermediate. This implies also a slightly different emitter in our case as compared to the base-induced chemiluminescence reaction in solution. Our experimental results on the photoluminescence of the main product of the solid-state thermochemiluminescence reaction (lophine) in different protonation states point towards the triplet emission of the deprotonated form of lophine as the emitter in case of the solid-state thermochemiluminescence reaction.

Comment: *A curiosity: were bis(adamantly)-1,2-dioxetane, benzoyl peroxide and rubrene endoperoxide heated as solids or as macroscopic crystals?*

Response to the comment: All of the mentioned compounds were heated as macroscopic crystals. This has been now clarified in the main text: **“To verify whether the solid-state thermochemiluminescence is limited to LHP or is a more common phenomenon, the emission from other organic peroxide classes, namely 1,2-dioxetanes, endoperoxides and aroyl peroxides in form of macroscopic crystals, was also recorded.”**

Comment: *Supporting Information:*

Chapter 1.3:

The protocol applied to isolate the decomposition products (lophine hydroperoxide dissolved in 50 mL of boiling toluene and stirred under reflux for more than 30 min) is significantly different from the protocol employed to record light emission by thermochemiluminescence (heating a macroscopic crystal). The decomposition products distribution could be different. If the authors work with 1 cm crystals, they could characterize the degradation products directly on the crystal after the light emission.

Response to the comment: We totally agree with the comment of the reviewer. Consequently, we performed UPLC-MS analysis of the thermally decomposed solid samples of lophine hydroperoxide. We further clarified the description of these experiments, which are displayed in the Supplementary Material, under section 1.3 (“Isolation of the decomposition products”) and Supplementary Figure 3. The results confirm that lophine is indeed the main reaction product and dibenzoylamidine as the side product #1. We were not able to identify the side product #2. The following text as added to detail these experiments:

“Analytical UHPLC analysis was also carried out on the remaining product after thermal decomposition of a solid sample of LHP using an Agilent Technologies 1290 Infinity II system equipped with an Agilent EclipsePlusC18 2.1 × 50 mm column with 1.8 μm silica beads together with an Agilent EclipsePlusC18 2.1 × 5 mm guard column with 1.8 μm silica beads. ...

Supplementary Figure 3. Analytical UHPLC chromatograms of lophine, lophine hydroperoxide (LHP) and the decomposition products of LHP.”

Comment: Chapter 1.7:

The activation parameters were derived from several isothermal chemiluminescence kinetics which were analyzed with Arrhenius and Eyring equations. Were the several isothermal chemiluminescence kinetics carried out at different temperatures? The method and the results should be illustrated in SI.

Response to the comment: We confirm that several isothermal chemiluminescence kinetics measurements were carried out at different temperatures. To clarify this point, we added a paragraph in section 1.7 of the Supplementary Material (“Optical

spectroscopy”) that now explains these experiments in greater detail and also contains the average kinetic constants calculated at each temperature:

“The kinetic trace of the thermochemiluminescence reaction follows first order kinetics ($R^2 > 0.99$). The rate constants were derived from this model (Supplementary Figure 15, Supplementary Table 4). The activation parameters (Supplementary Table 5) were also derived from the isothermal chemiluminescence kinetics which were analyzed with the Arrhenius equation (Supplementary Figure 16)

$$\ln(k) = \ln(A) - \frac{E_a}{R} \left(\frac{1}{T} \right)$$

and the Eyring equation:

$$\ln \frac{k}{T} = \frac{-\Delta H^\ddagger}{R} \cdot \frac{1}{T} + \ln \frac{k_B}{h} + \frac{\Delta S^\ddagger}{R}$$

Supplementary Figure 16. Arrhenius plot for the solid-state thermochemiluminescence reaction of LHP.

Supplementary Table 4. Rate constants of the solid state thermochemiluminescence reaction of LHP

T / K	k_{obs} / s^{-1}
423.15 ± 1	$8.27 \times 10^{-1} \pm 2.89 \times 10^{-2}$
413.15 ± 1	$4.03 \times 10^{-1} \pm 1.97 \times 10^{-2}$
403.15 ± 1	$1.58 \times 10^{-1} \pm 9.14 \times 10^{-3}$
393.15 ± 1	$6.35 \times 10^{-2} \pm 2.88 \times 10^{-3}$

$$383.15 \pm 1 \quad 1.44 \times 10^{-2} \pm 1.16 \times 10^{-3}$$

Supplementary Table 5. Activation parameter of the thermochemiluminescence reaction of LHP in solid state, and in chlorobenzene and N-methylpyrrolidinone solutions

	Solid	Chlorobenzene	NMP
E_a / kJ mol⁻¹	113.45 ± 4.61	88.41 ± 8.56	50.52 ± 0.30
ΔH^\ddagger / kJ mol⁻¹	114.61 ± 3.44	85.25 ± 8.58	47.41 ± 0.28
Δs^\ddagger / J K⁻¹ mol⁻¹	21.61 ± 0.82	-77.83 ± 14.93	-159.36 ± 3.14
ΔG^\ddagger / kJ mol⁻¹ (at 273 K)	108.71 ± 3.66	106.51 ± 12.66	90.94 ± 1.13

Comment: *Movie S7 (SEM micrographs of heated LHP crystals from room temperature to 160 °C) is missing.*

Response to the comment: The missing video was added in the revised version.

Response to the comments from Reviewer #3

Comments: ***Lophine** When refining this structure straight after extracting it from the embedded CIF file, the R factor will be 24.4% -- much higher than the reported 12.19%. I do not understand how this is possible. Diffraction of this material is reported to 0.83Å -- but in reality, the I/sigma drops below the 3-sigma line at around 1Å -- after that, the reflections are mostly noise: we are dealing with a low-resolution structure. The Rint is also very high, and it turns out that this material is twinned. Taking this twinning into account, the R factor will drop to around 11% (using all data). Cutting the data to exclude noise (OMIT -3 105) will lead to a (just) acceptable structure with an R1 of 8.54%. I attach my current refinement, which you may use as a starting point for your new refinement of this structure. There are no issues with the other two structures and I recommend publication of those.*

Response to the comments: We thank the reviewer for bringing to our attention the issue with the lophine structure. Although the structure of this compound has been previously reported, we re-determined it in our work for the purpose of identification of the product. Lophine crystallizes as very fine needles, most of which are twinned. In our first submission, the structure was determined from a very thin needle-like crystals, and this resulted in poor diffraction. We have now grown a new batch of crystals and we collected data from a crystal which is not twinned, although the weak diffraction could not be avoided. As a result, the new crystal structure is of better quality. The new crystallographic data have been added to Supplementary Table 1. The new data were deposited with the CCDC and were assigned a deposition number 1885196. The CIF, structure factors (FCF file) and CheckCIF report for the new structure are included with this submission as Supplementary Material.

REVIEWERS' COMMENTS:

Reviewer #1 (Remarks to the Author):

After careful reading of the revised manuscript and the author's response to the referees' comments, I consider that the work by Prof. Naumov and co-workers should be published in Nature Comm., in its current state.

Reviewer #2 (Remarks to the Author):

The authors have clarified many points both explaining better some concepts and carrying out further experiments/measurements.

In my opinion, the work is now convincing and I recommend its publication in Nature Communications.

Arianna Quintavalla

Reviewer #3 (Remarks to the Author):

There is 1 Structure in this paper. We examined this file: 1885196

Thank you for going through the trouble and collecting a new dataset here. This structure is indeed much better, and I recommend publication without further hesitation.

Response to the queries and list of changes made to the final version of the manuscript NCOMMS-18-31289A:

Response to the reviewers' comments:

Comment: Reviewer #1 (Remarks to the Author):

After careful reading of the revised manuscript and the author's response to the referees' comments, I consider that the work by Prof. Naumov and co-workers should be published in Nature Comm., in its current state.

Response: We thank the Reviewer for their time and effort with the comments.

Comment: Reviewer #2 (Remarks to the Author):

The authors have clarified many points both explaining better some concepts and carrying out further experiments/measurements. In my opinion, the work is now convincing and I recommend its publication in Nature Communications.

Arianna Quintavalla

Response: We thank the Reviewer for their time and effort with the comments.

Comment: Reviewer #3 (Remarks to the Author):

There is 1 Structure in this paper. We examined this file: 1885196

Thank you for going through the trouble and collecting a new dataset here. This structure is indeed much better, and I recommend publication without further hesitation.

Response: We thank the Reviewer for their time and effort with the comments.